# HHD-Ethiopic
## A Historical Handwritten Dataset for Ethiopic OCR with Baseline Models and Human-level Performance

**Birhanu Hailu Belay**[*]   **Isabelle Guyon**[+*‡]   **Tadele Mengiste**[†]   **Bezawork Tilahun**[†]
**Macus Liwicki**[‖]   **Tesfa Tegegne**[†]   **Romain Egele**[*]   **Tsiyon Worku**[†]
[*]LISN, Université Paris-Saclay, France    [+] Google Brain, USA    [‡]ChaLearn,USA
[†]Bahir Dar Institute of Technology, Ethiopia    [‖]Luleå University of Technology, Sweden
birhanu-hailu.belay@upsaclay.fr
guyon@chalearn.org, marcus.liwicki@ltu.se, romain.egele@inria.fr
{tadele.mengiste,bezawork.tilahun,tesfa.tegegne,tsiyon.worku}@bdu.edu.et

## Abstract

This paper introduces HHD-Ethiopic, a new OCR dataset for historical handwritten Ethiopic script, characterized by a unique syllabic writing system, low resource availability, and complex orthographic diacritics. The dataset consists of roughly 80,000 annotated text-line images from 1700 pages of $18^{th}$ to $20^{th}$ century documents, including a training set with text-line images from the $19^{th}$ to $20^{th}$ century and two test sets. One is distributed similarly to the training set with nearly 6,000 text-line images, and the other contains only images from the $18^{th}$ century manuscripts, with around 16,000 images. The former test set allows us to check baseline performance in the classical IID setting (Independently and Identically Distributed), while the latter addresses a more realistic setting in which the test set is drawn from a different distribution than the training set (Out-Of-Distribution or OOD). Multiple annotators labeled all text-line images for the HHD-Ethiopic dataset, and an expert supervisor double-checked them. We assessed human-level recognition performance and compared it with state-of-the-art (SOTA) OCR models using the Character Error Rate (CER) and Normalized Edit Distance(NED) metrics. Our results show that the model performed comparably to human-level recognition on the $18^{th}$ century test set and outperformed humans on the IID test set. However, the unique challenges posed by the Ethiopic script, such as detecting complex diacritics, still present difficulties for the models. Our baseline evaluation and HHD-Ethiopic dataset will stimulate further research on tailored OCR techniques for the Ethiopic script. The HHD-Ethiopic dataset and the code are publicly available at `https://github.com/bdu-birhanu/HHD-Ethiopic`.

## 1   Introduction

The gathering of historical knowledge heavily relies on analyzing digitized historical documents [25]. In order to process a large number of these document images, automated tools that can convert images of the original handwritten documents into its digital format (e.g., with Unicode or ASCII texts ) are necessary [7, 48]. One such tool is Optical Character Recognition (OCR), which enables computers to extract textual information contained in images to then provide editing, translation, or search capabilities [13, 46]. OCR systems often face difficulty in accurately recognizing historical

Submitted to the 37th Conference on Neural Information Processing Systems (NeurIPS 2023) Track on Datasets and Benchmarks. Do not distribute.

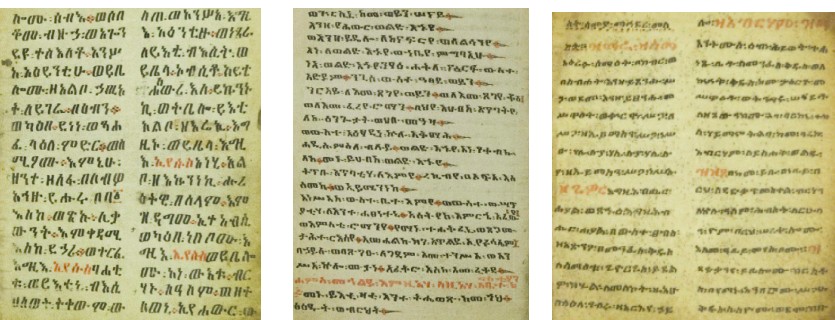

Figure 1: Sample historical handwritten document image from HHD-Ethiopic dataset: two-column $19^{th}$-century manuscript (left), one-column $20^{th}$-century manuscript (middle), two-column $18^{th}$-century manuscript (right).

|   | 0 | 1 | 2 | 3 | 4 | 5 | 6 |
|---|---|---|---|---|---|---|---|
| 0 | ሀ hâ | ሁ hu | ሂ hí | ሃ ha | ሄ hē | ህ hi | ሆ ho |
| 1 | ለ lâ | ሉ lu | ሊ lí | ላ la | ሌ lē | ል li | ሎ lo |

Figure 2: The first two top rows of Fidel-Gebeta (the row-column matrix structure of Ethiopic characters): The first column shows the consonants, while the following columns (1-6) illustrate syllabic variations (obtained by adding diacritics or modifying parts of the consonant, circled in color). These modifications results a complex and distinct characters having similar shape, which making them challenging for machine learning models (see Appendix B)

documents, particularly those written in Ethiopic scripts, due to a shortage of suitable datasets for training machine learning models and the unique complexities of orthography [8, 34]. Typical historical handwritten Ethiopic manuscripts from different centuries are displayed in Figure 1.

The Ethiopic script, also known as the Abugida, Ge'ez, or Amharic script, is one of the oldest writing systems in the world, with a history dating back to the $4^{th}$ century AD [22]. It is used to write several languages in Ethiopia and Eritrea, including Amharic, Tigrinya, and Ge'ez. The script has a unique syllabic writing system and is written from left to right. It contains about 317 graphemes, including 231 basic characters arranged in a 33 consonants by 7 vowels matrix, one special $(1 \times 7)$ character, 50 labialized characters, 9 punctuation marks, and 20 numerals. The script's complexity is increased by the presence of diacritical marks, which are used to indicate vowel length, tone, and other phonological features. [2, 32, 30] (see Appendix B). The first two consonant Ethiopic characters and their corresponding vowels formation is shown in Figure 2.

The Ethiopian National Archive and Library Agency (ENALA) has collected numerous non-transcribed historical Ethiopic manuscripts from various sources, covering different periods starting from the $12^{th}$ century [49]. These documents are manually cataloged and some are digitized and stored as scanned copies. They contain valuable information about Ethiopian cultural heritage and have been registered in UNESCO's Memory of the World program [9, 35]. The manuscripts are mainly written in Ge'ez and Amharic languages, which share the same syllabic writing system.

To address the scarcity of suitable datasets for machine learning tasks in historical handwritten Ethiopic text-image recognition, we aim to prepare a new dataset that can advance research on the Ethiopic script and facilitate access to knowledge from these historical documents by various communities, including paleographers, historians, librarians, and researchers.

The main contributions of this paper are stated as follows.

- We introduce the first sizable dataset for historical handwritten Ethiopic text-image recognition, named HHD-Ethiopic.

- We evaluate an independent human-level performance from multiple participants in historical handwritten text-image recognition, providing a baseline for comparison with machine learning models.

- We evaluate several state-of-the-art Transformer, attention, and Connectionist Temporal Classification (CTC)-based methods.

- We compare the recognition performance of the machine learning model with human-level performance in predicting the sequence of Ethiopic characters in text-line images, supported by examples.

The rest of the paper is organized as follows: Section 2 reviews the relevant methods and related works. Settings of human-level recognition performance and OCR models are described in section 3. Section 4 presents results obtained from the experiment and comparative analysis between the model and human-level recognition performance. Finally, in Section 5, we conclude and suggest directions for future works.

## 2 Related work

In this section, we briefly review related work in optical character recognition and highlight challenges we are facing in OCR of historical Ethiopic manuscripts.

### 2.1 Optical character recognition

Machine Learning techniques have been extensively applied to the problem of optical character recognition, see [11, 48, 51, 12, 50] for a review. This has been facilitated by the public availability of a multitude of datasets for various document image analysis tasks, in a variety of scripts: Among these, we can mention IAM-HistDB[19], DIDA [24], IMPACT [37], GRPOLY-DB [20], DIVA-HisDB [44], ICDAR-2017 Dataset [39], SCUT-CAB [13] and HJDataset [40] as examples of historical and handwritten datasets. There are other datasets that can be used for printed and scene text-image recognition, including the ADOCR database [8], OmniPrint datasets [45], UHTelPCC [23], COCO dataset [47], and TextCaps [43], in addition to the historical and handwritten datasets mentioned previously.

Nowadays, segmentation-free OCR approaches [3, 36, 51] based on CTC [7, 15, 11, 31, 48, 41] attention mechanisms [27, 38, 42, 50], and transformer-based models [5, 18, 26, 33] have become a popular choice among researchers and are widely used for text-image recognition (in both well-known and low-resourced scripts), as opposed to the traditional segmentation-based OCR approaches. Researchers have reported remarkable recognition performance using these approaches for a wide range of scripts, encompassing everything from historical to modern [5, 28], and from handwritten to machine-printed [9]. Consequently, several OCR applications have been developed that perform exceptionally well for high-resource and well-known scripts. However, many of these applications have not been assessed for their ability to recognize text in historical handwritten manuscripts and missing these potential benefits, especially in the case Ethiopic manuscripts. In the following sections, we briefly discuss the features of historical Ethiopic manuscripts and the challenges of text-image recognition in ancient Ethiopic manuscripts.

### 2.2 Features of historical Ethiopic manuscripts

There are various collections of ancient Ethiopic manuscripts in museums and libraries in Ethiopia and other countries. For example, the ENALA collection contains 859 manuscripts, the Institutes of Ethiopian Studies has 1500 manuscripts [35, 1], and the collections in Rome (Biblioteca Apostolica Vaticana), Paris (Bibliothèque nationale de France), and London (British Library) contain a total of 2700 manuscripts [35]. These manuscripts were typically written on a material called Brana, which could vary in quality depending on the intended purpose or function of the book [29, 35]. Black and red were the most commonly used inks, with black reserved for the main text and red reserved

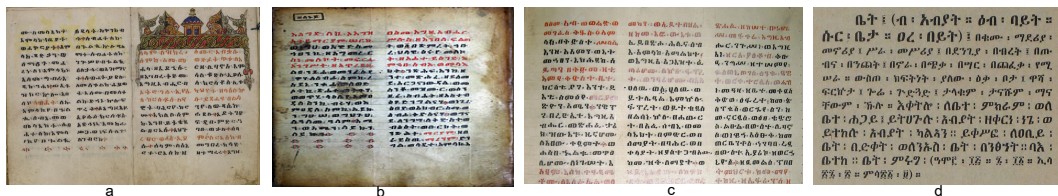

Figure 3: Examples of historical Ethiopic Manuscripts: (a) Two-column writing in liturgical books with decorated heading[1], (b) Two-column writing in liturgical books without decoration[2], (c) Three-column writing in the Synaxarion[3], (d) One column for Psalms and prayer books[4]. The Ethiopic script is written and read in the same direction as English, from left to right and top to bottom.

for religious headings and names of significance. Figure 3 shows examples of historical Ethiopic manuscripts.

The manuscript layout can also vary and include formats such as three columns in the Synaxarion, one column for Psalms and prayer books, and two columns in liturgical books [6, 35]. The materials used for writing, including the pen and ink and the writing style, can also vary depending on the time period and region in which the manuscripts were produced. The use of punctuation marks is also very irregular (see Appendix B, Figure 10 for an extended discussion).

Historical documents, such as Ethiopic manuscripts, often have artifacts like color bleed-through, paper degradation, and stains, making them more challenging to work with than contemporary, well-printed documents [17]. Some major challenges in recognizing historical Ethiopic manuscripts include: (i) the complexity of character sets and writing system, which consists of over 317 distinct but similar-looking indigenous characters (see Figure 2 and details are given in Appendix B); (ii) variations in writing styles, including handwriting and punctuation, which can vary greatly among individuals and over time, affecting model accuracy; and (iii) a shortage of labeled data for training machine learning algorithms for Ethiopic script recognition.

Therefore, in this paper, we aim to tackle the challenges in recognizing the Ethiopic script by creating a new dataset called HHD-Ethiopic which is composed of manuscripts dating from the $18^{th}$ to $20^{th}$ centuries. We also conduct experimental evaluations to showcase the usefulness of the HHD-Ethiopic dataset for historical handwritten Ethiopic script recognition and compare the performance of both human and machine predictions.

## 3 Dataset and baseline methods

In this section, we provide an overview of our work, focusing on two key aspects: the detailed characteristics of our new dataset (subsection 3.1) and the benchmark methods employed. Our dataset, comprehensively outlined, includes essential details such as size, composition, data collection, and annotation process. It serves as a valuable resource for evaluating historical handwritten Ethiopic OCR. Additionally, we present the benchmark methods, including human-level recognition performance and baseline OCR models (subsection 3.2).

### 3.1 HHD-Ethiopic dataset

The HHD-Ethiopic dataset consists of 79,684 text-line images with their corresponding ground-truth texts that are extracted from 1,746 pages of Ethiopic manuscripts dating from $18^{th}$ to $20^{th}$ centuries. The dataset includes 306 unique characters (including one blank token), with the shortest text comprising two characters and the longest containing 46 characters. These 306 characters are

---

[1] https://expositions.nlr.ru/eng/ex_manus/efiopiia/efiopiia_letter.php

[2] https://upload.wikimedia.org/wikipedia/commons/2/2f/Sample_of_Ge%27ez_writing.jpg

[3] https://elalliance.files.wordpress.com/2013/11/world-history2.jpg

[4] https://www.w3.org/TR/elreq/images/kwk-mashafa-sawasew-page-268-typeface-change-for-emphasis.jpg

Table 1: Summary of the training and test text-line images

| Type-of-data | Pub-date-of-manuscript | #text-line images | remark |
|---|---|---|---|
| Training-set | 90% of (A+B+C) | 57,374 | real |
| Test-set-I (IID) | 10% of (A+B+C) | 6,375 | real |
| Test-set-II (OOD) | 100% of (D) | 15,935 | real |

A= Unknown pub. date, B= $20^{th}$ century, C= $19^{th}$ century, D= $18^{th}$ century manuscript

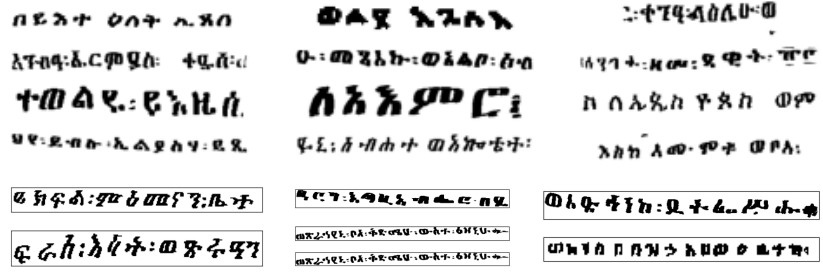

Figure 4: Sample historical handwritten Ethiopic text-line images from HHD-Ethiopic

not distributed equally; some occur more frequently due to the nature of the script, being widely used in the writing system. For example characters such as ወ, ነ, ስ, በ, ት, ይ, ም, ላ, ር, ተ, ም, ብ, ከ, ል, etc are among the most frequent characters, whereas characters like ፐ, ፓ, ዣ, ዥ, ኺ, ጷ, ፒ, ጁ, ሯ, ፒ, etc are notably infrequent, occurring almost below a count of 10. In response to this issue of underrepresentation, we have generate a separate synthetic text-line images from these characters (see the Appendix section C.4 for an extended discussion)

The training set includes text-line images from recent manuscripts, primarily from the $19^{th}$ and $20^{th}$ centuries. We created two test set: the first one consists of 6375 images that are randomly selected using a sklearn train/test split protocols[5], from a distribution similar to the training set, specifically from $19^{th}$ and $20^{th}$ century books. The second one, with 15,935 images, is drawn from a different distribution and made of $18^{th}$ century manuscripts (see Table 1 for the splitting processes and size of the each set). The goal of the first test set is to evaluate the baseline performance in the IID (Independently and Identically Distributed) setting, while the second test aims to assess the model's performance in a more realistic scenario, where the test set is OOD (Out-Of-Distribution) and different from the training set.

To perform preprocessing and layout analysis tasks, such as text-line segmentation, we utilized the OCRopus[6] framework. For text-line image annotation, we developed a simple tool with a graphical user interface, which displays an image of a text-line and provides a text box for typing and editing the corresponding ground-truth text. Additionally, we employed this tool to collect predicted text during the evaluation of human-level performance.

A team of 14 people participated in creating the HHD-Ethiopic datasets, with 12 individuals tasked with labeling and the remaining two individuals responsible for reviewing and ensuring the accuracy of the alignment between the ground-truth text and text-line images, making any necessary corrections as needed. To ensure the accuracy of the annotations, participants were provided with access to reference materials for the text-lines, and all of them were familiar with the characters in the Ethiopic script. Table 1 and Figure 4 provide a summary of the dataset and show sample text-line images of the HHD-Ethiopic dataset, respectively (see Appendix C.3 for an extended discussion).

---

[5] https://scikit-learn.org/stable/modules/generated/sklearn.model_selection.train_test_split.html
[6] https://github.com/ocropus/ocropy

### 3.2 Settings for human-level performance and baseline models

To establish a baseline for evaluating the performance of models on the HHD-Ethiopic OCR dataset, we propose two approaches: (i) **Human-level performance** and (ii) **Sequence-to-sequence models**.

The human-level performance serves as a benchmark for evaluating and comparing the recognition performance of machine learning models on historical handwritten Ethiopic scripts and provides insights for error analysis. To calculate the human-level recognition performance, 13 independent annotators were hired and divided into two groups. It is important to note that these individuals are different from those mentioned in section 3.1. The first group transcribed text-line images from the first test set, which consisted of 6375 randomly selected images from the training set. The second group transcribed the second test set of 15935 images from the $18^{th}$ century. Each text-line image was predicted by multiple people (i.e nine for Test-set-II and four for Test-set-II). The annotators were already familiar with the Ethiopic script, and they were explicitly instructed to carry out the task without using any references. The predicted texts by each annotator, along with comprehensive details of the data collection and annotation process, is documented as metadata for future reference.

The second reference point involves various state-of-the-art OCR models, which includes CTC, attention and transformer-based methods. The CTC-based models employ a combination of Convolutional Neural Networks (CNN) and Bidirectional Long Short-Term Memory (Bi-LSTM) as an encoder and CTC as a decoder, and is trained end-to-end with and without an Attention mechanism (see Appendix C for an extended discussion). In addition, for the attention-based baseline, we employ ASTER [42], and for the transformer-based baselines, we utilize the ABINet[18] and TrOCR [26].

Moreover, we use Bayesian optimization (see e.g., [4, 16] for a review) to optimize the hyperparameters of the CTC-based models. Optimizing hyperparameters involves finding an optimal setting for the model hyperparameters that could result in the best generalization performance, without using test data. Considering the trade-offs between model performance and computational cost, we use a small subset of the training set to optimize the hyperparameters of models (see, e.g.,[10] for a review), and then train the model on the full training set using the optimal hyperparameter settings.

We used the Character Error Rate (CER) [7, 21] and Normalized Edit Distance (NED)[14] as our evaluation metric for both the OCR models and human-level recognition performances (see appendix C, equation 3 and 4 for extended discussion).

## 4 Experimental results

Our objective is to perform a fair comparison between human and machine performance on historical handwritten Ethiopic scripts recognition task. This comparison is intended to showcase the utility and value of our new HHD-Ethiopic dataset, evaluate human recognition capabilities, and highlight any advancements made by baseline OCR methods.

### 4.1 Human-level performance

As previously discussed in Section 3.1, the ground-truth text was annotated by multiple people and double-checked by supervisors who were familiar with Ethiopic scripts. For this phase, new annotators who were also familiar with Ethiopic characters were selected and instructed not to use any reference materials. The reviewer of both the training and test sets was permitted to use reference materials. However, in contrast to the training set, the test sets were reviewed by an expert in historical Ethiopic documents.

To measure the human-level recognition performance, multiple annotators were asked to predict the text in the images and then their character recognition rates were recorded. The best annotator on Test-set-I scored a CER of 25.39% and an NED of 23.78% on Test-set-I, and a CER of 33.20% and an NED of 30.73% on Test-set-II. In contrast, the average human-level recognition performance was a CER of 30.46% and an NED of 26.32% on Test-set-I, and a CER of 35.63% and an NED of 38.59% on Test-set-II. We used the best human-level recognition performance as a baseline for

Table 2: The human-level recognition performance in Character Error Rates (CER) and Normalized Edit Distance(NED)

| Type-of-test data | Year-of-Pub | Annotator-ID | CER | NED |
|---|---|---|---|---|
| IID | $19^{th}$ & $20^{th}$ | Annot-I | 29.02 | 27.67 |
| | | Annot-II | 27.87 | 25.89 |
| | | Annot-III | 29.93 | 28.16 |
| | | Annot-IV | 29.16 | 27.80 |
| | | Annot-V | 26.56 | 24.56 |
| | | Annot-VI | **25.39** | **23.78** |
| | | Annot-VII | 29.26 | 28.08 |
| | | Annot-VIII | 25.95 | 24.78 |
| | | Annot-IX | 51.03 | 25.46 |
| OOD | $18^{th}$ | Annot-X | **33.20** | **30.77** |
| | | Annot-XI | 54.33 | 52.20 |
| | | Annot-XIII | 39.96 | 35.90 |
| | | Annot-XIV | 45.06 | 39.89 |

comparison with SOTA machine learning models' performance throughout this paper. Table 2, shows the human-level recognition performance on both test sets, based on assessments from nine annotators on Test-set-I and four on Test-set-II.

## 4.2 Baseline OCR models

This section presents the results obtained from the experimental setups detailed in Section 3. Firstly, we present the results of the CTC-based OCR models previously proposed for Amharic script recognition [7, 9], followed by the results of other state-of-the-art models [15, 18, 26, 41, 42] validated in Latin and/or Chinese scripts.

The experiments conducted using the CTC-based models previously proposed for Amharic script were categorized into four groups:

- **HPopt-Plain-CTC**: plain-CTC (optimized hyper-parameters)
- **Plain-CTC**: Plain-CTC
- **HPopt-Attn-CTC**: Attention-CTC (optimized hyper-parameters)
- **Attn-CTC**: Attention-CTC

In all the CTC-based setups, to minimize computational costs during training, we resized all the text-line images to 48 by 368 pixels. We used 10% of the text-line images randomly drawn from the training set for validation. As previously discussed, in Section 3, we have two test sets: (i) Test-set-I, which includes 6375 text-line images randomly selected from $19^{th}$, $20^{th}$ century manuscripts and other manuscripts with unknown publication dates, and (ii) Test-set-II, a text-line images that are drawn from a different distribution other than the training set, which includes 15935 text-line images from $18^{th}$ century Ethiopic manuscripts only. The HPopt-Attn-CTC baseline model achieved the best CER of 16.41% and 28.65% on Test-set-I and Test-set-II, respectively (see Table 3 for details).

The results depicted in Figure 5 demonstrate that the CTC-based OCR models outperform human-level performance on Test-set-I in all configurations. However, only the HPopt-Attn-CTC model can surpass human-level performance, while the other configurations achieve comparable or worse results compared to human recognition on Test-set-II. Test-set-I was randomly selected from the training set, while Test-set-II consisted of $18^{th}$ century manuscripts and represented out-of-distribution data. This disparity in performance is to be expected, as machine learning models typically perform better on samples that are independently and identically distributed rather than those in an out-of-distribution setting. The repeat experiments aimed to capture the variability in the performance of the models due to random weight initialization and sample order.

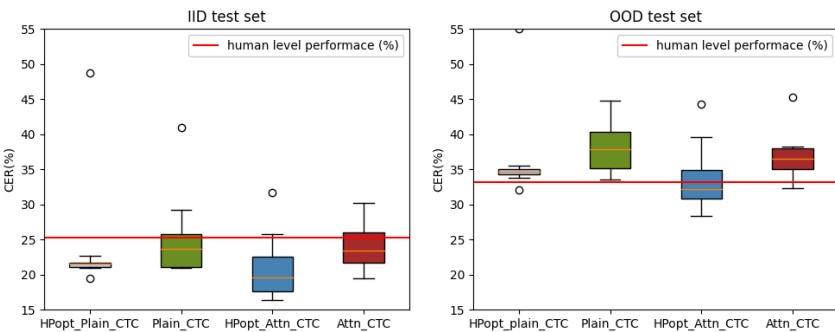

Figure 5: Box Plot comparison of variance in the recognition performance of CTC-based models and human level performance from ten experiments with varying random weight initialization and training sample orders on Test-set-I (IID) (left) and Test-set-II(OOD) (right). The results demonstrate that HPopt-attn-CTC outperforms all other CTC-based methods and surpassing human-level recognition on both test sets. The second group of models, being complex, was run through individual experiments. Instead of utilizing a Box Plot, a learning curve is provided (see Appendix C.2 for an extended discussion)

HPopt-plain-CTC exhibits consistent variability across the 10 experiments due to the benefits of hyper-parameter optimization and a simplified architecture without attention mechanisms. The systematic fine-tuning of hyper-parameters, coupled with a simpler model architecture, resulted in stable and predictable performance throughout the experiments. In contrast, HPopt-attn-CTC achieved the lowest error despite some variability in certain trials, demonstrating its robustness across ten trials (see Table 3). The optimized hyperparameter configuration significantly improved recognition accuracy compared to non-optimized settings on both test sets, highlighting the importance of hyperparameter tuning for superior performance beyond relying solely on prior knowledge or trial-and-error approaches.

The second category of baseline OCR models assessed using our HHD-Ethiopic dataset comprises state-of-the-art models, including CRNN [41], ASTER [42], ABINet [18], SVTR [15], and TrOCR [26]. Considering our available computing resources, except for the TrOCR model, which was trained with few iterations, all other models were trained for 25 epochs. The learning curve, which illustrates the recognition performance using a CER metric on the IID and OOD test sets, is presented in Appendix Figure 12. In this group, the SVTR and ABINet models achieved the highest performance, with both models showing nearly equivalent results within a 1% difference during evaluation. As shown in Table 3, compared to the CTC-based models, the attention and transformer-based models exhibit larger number of parameter (see Appendix C for an extended discussion).

Based on Figure 6 and our experimental observations, we observed distinct error patterns between humans and models: both exhibit substitution errors, but the model tends to make a higher number of insertions and deletions. This highlights the imperfection of the baseline OCR models in terms of sequence alignments. Furthermore, our study found that the evaluated baseline OCR models were highly effective, surpassing human-level recognition performance on Test-set-I. However, only a few models achieved better recognition performance on Test-set-II. Compared to other methods, the HPopt-Attn-CTC model has achieved the best recognition accuracy on both datasets.

The baseline models evaluated in this study comprise CTC-based models previously proposed for the Amharic script, alongside five state-of-the-art attention and transformer-based models validated using English and Chinese scripts. These models could serve as references for evaluating the effectiveness of advanced models in recognizing historical handwritten Ethiopic scripts. Each of the CTC-based models previously proposed for Amharic script underwent ten experiments. In contrast, the other models, although trained for only single experiments and fewer epochs, achieved comparable outcomes. In addition, among the CTC-based models, the optimized hyperparameters model demonstrates

---

[7]https://matplotlib.org/stable/api/_as_gen/matplotlib.pyplot.boxplot.html

Table 3: A summary of baseline models and their recognition performance on Test-set-I (IID, 6k) and Test-set-II (OOD, 16k) using CER and NED. The table includes model parameters measured in millions (M) and presents the lower and upper quartiles, denoted as [val$^-$, val$^+$], obtained from multiple experiments.

| Methods | #Model-Parms | Type-of-test data | [val$^-$, val$^+$] | CER | NED |
|---|---|---|---|---|---|
| Plain-CTC[7] | 2.5M | IID | [21.05, 25.80] | 20.88 | 19.09 |
| | | OOD | [35.15, 40.38] | 33.56 | 31.9 |
| Attn-CTC [9] | 1.9M | IID | [21.05, 26.01] | 19.42 | 21.01 |
| | | OOD | [35.00, 37.94] | 33.07 | 32.92 |
| HPopt-Plain-CTC | 4.5M | IID | [21.02, 21.73] | 19.42 | 17.77 |
| | | OOD | [34.32, 34,98] | 32.01 | 29.02 |
| HPopt-Attn-CTC | 2.2M | IID | [17.55, 22.56] | **16.41** | **16.06** |
| | | OOD | [30.79, 34,88] | **28.65** | **27.37** |
| TrOCR[26] | 333.9M | IID | - | 35.0 | 33.0 |
| | | OOD | - | 45.0 | 43.87 |
| CRNN [41] | 8.3M | IID | - | 21.04 | 21.01 |
| | | OOD | - | 29.86 | 29.29 |
| ASTER [42] | 27M | IID | - | 24.43 | 20.88 |
| | | OOD | - | 35.13 | 30.75 |
| SVTR [15] | 6M | IID | - | 19.78 | 17.98 |
| | | OOD | - | 30.82 | 28.00 |
| ABINet [18] | 23M | IID | - | 21.49 | 18.11 |
| | | OOD | - | 32.76 | 28.84 |
| Human-performance | | IID | [26.56, 29.26] | 25.39 | 23.78 |
| | | OOD | [38.27, 47.38] | 33.20 | 30.77 |

- denotes no lower/upper quartiles due to model complexity; single experiment with ASTER, CRNN, SVTR, ABINet, and TrOCR models

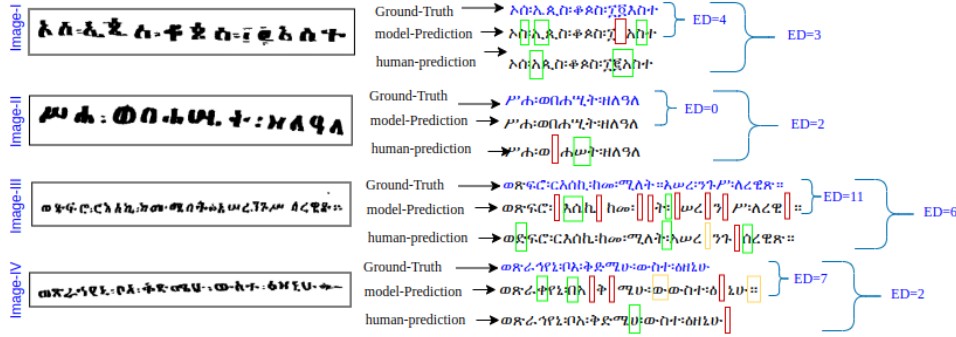

Figure 6: Sample human-machine recognition errors per text-line image from the Test-set-I. Deleted characters are marked in red, while substituted and inserted characters are marked by green and yellow boxes, respectively. The inner ED denotes the Edit distance between the ground-truth and model prediction, while the outer ED denotes ground-truth to human prediction Edit distance.

superior performance, benefiting from fine-tuning and reduced overfitting. The reported results and dataset serve as a benchmark for future research in machine learning, historical document image analysis, and recognition, while the analysis of human-level recognition performance enhances our understanding of the dataset.

# 5    Conclusion

In this paper, we presented a novel dataset for text-image recognition research in the field machine learning and historical handwritten Ethiopic scripts. The dataset comprises 79,684 text-line images obtained from manuscripts ranging from the $18^{th}$ to $20^{th}$ centuries and includes two test sets for evaluating OCR systems in both the IID (Independent and Identically Distributed) and OOD (Out-

of-Distribution) settings. We provided human-level performance and baseline results using CTC, attention and transformer based models to aid in the evaluation of OCR systems. To the best of our knowledge, this is the first study to offer a sizable historical dataset with human-level performance in this domain.

In addition to the human-level performance, we demonstrated the use of our dataset in addressing the problem of text-image recognition. We evaluated it using previously proposed models for Amharic script and state-of-the-art models validated with Latin and Chinese scripts. We evaluated their performance using the Character Error Rate (CER)and Normalized Edit Distance (NED). Our experiments demonstrate that both the trained SOTA methods and the smaller networks yield comparable results. Notably, the SOTA models produce equivalent outcomes even with fewer and smaller iterations, but larger parameter size. The smaller networks requires multiple experiments, making them suitable for low-resource computing infrastructure while still achieving comparable results.

The dataset and source code can be accessed at `https://github.com/bdu-birhanu/HHD-Ethiopic`, serving as a benchmark for machine learning and historical handwritten Ethiopic OCR research in low-resource settings. One limitation of our work is the scarcity of rare characters within the dataset. To tackle this limitation, we generate synthetic text-line images for the less frequent characters. However, our models have not been trained extensively using a larger synthetic dataset due to constraints on computational resources. To address this, future work includes expanding the dataset, and incorporating language models and contextual information for improved recognition. Additionally, we aim to refine the baseline models and conduct further experiments to enable a more systematic and conclusive evaluation of the different methods.

## Acknowledgments and Disclosure of Funding

This work was partially supported by ChaLearn, the ANR ( L'agence Nationale de la Recherche) under the Chair for AI Democratization , project grant number ANR-19-CHIA-0022 and ICT4D research center of Bahir Dar Institute of Technology. We are also grateful to the the Ethiopian National Archive and Library Agency (ENALA) staffs who provided valuable assistance with data collection, and allowing us access to necessary documents, as well as to Tariku Adane, Gizaw Wakjira, and Lemma Kassaye for their help with data collection approval and validation.

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
