# Supplementary file for "HHD-Ethiopic: A Historical Handwritten Dataset for Ethiopic OCR"

**Birhanu Hailu Belay**[*] **Isabelle Guyon**[+*‡] **Tadele Mengiste**[†] **Bezawork Tilahun**[†]
**Macus Liwicki**[‖] **Tesfa Tegegne**[†] **Romain Egele**[*] **Tsiyon Worku**[†]
[*]LISN, Université Paris-Saclay, France [+] Google Brain, USA [‡]ChaLearn,USA
[†]Bahir Dar Institute of Technology, Ethiopia [‖]Luleå University of Technology, Sweden
birhanu-hailu.belay@upsaclay.fr
guyon@chalearn.org, marcus.liwicki@ltu.se, romain.egele@inria.fr
{tadele.mengiste,bezawork.tilahun,tesfa.tegegne,tsiyon.worku}@bdu.edu.et

## 1 Appendix

This appendix comprises three sections: dataset documentation, Ethiopic writing system, and dataset preparation and baseline model training details. The dataset documentation outlines its composition, preprocessing steps, recommended use-case of distribution of the HHD-Ethiopic dataset, and author statement. The Ethiopic writing system section explores its historical significance and script structure. Lastly, the dataset and baseline training process section offers insights into dataset preparation and baseline model training strategies.

## A Dataset documentation for HHD-Ethiopic

To prepare this dataset documentation, we use a datasheet [11] for dataset guideline. This documentation consists of the motivation behind the dataset, its composition, the process of collection, recommended use cases, as well as information on processing, cleaning, labeling, distribution (including hosting, licensing), and maintenance. This documentation also includes author statements.

### A.1 Motivation

**For what purpose was the dataset created?** Was there a specific task in mind? Was there a specific gap that needed to be filled? Please provide a description.

The dataset targets the challenges of the indigenous Ethiopic script, addressing its scarcity of resources. It serves as a valuable asset for researchers and developers, facilitating advancements in OCR technology specifically for historical handwritten Ethiopic recognition. Unlike well-studied scripts like Latin, it bridges the gap and enables accurate recognition of Ethiopic text in historical documents using machine learning approaches.

**Who created this dataset** (e.g., which team, research group) and on behalf of which entity (e.g., company, institution, organization)?

HHD-Ethiopic dataset is created primarily by the LISN lab at University of Paris-Saclay and ICT4D research center at Bahir Dar Institute of Technology, in collaboration with other researchers from Lulea Technology University.

**Who funded the creation of the dataset?** If there is an associated grant, please provide the name of the grantor and the grant name and number.

Submitted to the 37th Conference on Neural Information Processing Systems (NeurIPS 2023) Track on Datasets and Benchmarks. Do not distribute.

The dataset creation received funding from ChaLearn and the ICT4D research center of Bahir Dar Institute of Technology. Findings, and/or recommendations expressed in this material are solely those of the author/s and do not necessarily represent the views of ChaLearn or ICT4D.

**Any other comments?** No.

## A.2 Composition

**What do the instances that comprise the dataset represent (e.g., documents, photos, people, countries)?** Are there multiple types of instances (e.g., movies, users, and ratings; people and interactions between them; nodes and edges)? Please provide a description.

The HHD-Ethiopic dataset is an OCR dataset that consists of text-line images extracted from historical handwritten Ethiopic manuscript and there corresponding ground truths text, sample images and their corresponding ground truth texts are shown Figure 14

**How many instances are there in total (of each type, if appropriate)?**

The HHD-Ethiopic dataset comprises 79,684 text-line images accompanied by their respective ground-truth texts. These images are extracted from a collection of 1,746 pages of Ethiopic manuscripts dating from the $18^{th}$ to the $20^{th}$ centuries. The dataset is divided into a training set, containing 57,374 text-line images, and two distinct Test sets. One Test set, that consists 6,375 images, is randomly sampled from the training set, while the other is exclusively prepared from the $18^{th}$ century Ethiopic manuscripts and includes about 15,935 text-line images along with their corresponding ground-truth texts ( details are provided in the main paper).

**Does the dataset contain all possible instances or is it a sample (not necessarily random) of instances from a larger set?** If the dataset is a sample, then what is the larger set? Is the sample representative of the larger set (e.g., geographic coverage)? If so, please describe how this representativeness was validated/verified. If it is not representative of the larger set, please describe why not (e.g., to cover a more diverse range of instances, because instances were withheld or unavailable).

HHD-Ethiopic, is a historical handwritten dataset between the $18^{th}$ and $20^{th}$ centuries. It is a sample of instances from that time period and includes 306 out of 317 frequently used characters in the Ethiopian writing system.

**What data does each instance consist of? "Raw" data (e.g., unprocessed text or images) or features?** In either case, please provide a description.

Each instance in the training set consists of text-line images and their corresponding ground-truth text. The test set, on the other hand, includes raw human-level prediction texts from 13 independent annotators which we use as a baseline to compare the human-level performance with OCR models in this paper

**Is there a label or target associated with each instance?** If so, please provide a description.

Yes, there is a ground-truth text for each text-line image.

**Is any information missing from individual instances?** If so, please provide a description, explaining why this information is missing (e.g., because it was unavailable). This does not include intentionally removed information, but might include, e.g., redacted text.

No, everything is included.

**Are relationships between individual instances made explicit (e.g., users' movie ratings, social network links)?** If so, please describe how these relationships are made explicit.

The relationships between individual instances in the text-line image dataset are not explicitly defined, as each image is formed from a sampled set of 306 Ethiopic characters rather it may have indirect/inferred connection.

**Are there recommended data splits (e.g., training, development/validation, testing)?** If so, please provide a description of these splits, explaining the rationale behind them.

The HHD-Ethiopic dataset is split into first into training, and testing. The training set includes text-line images from the $19^{th}$ and $20^{th}$ centuries. A validation set is then randomly sampled as 10% of the training set. Two test sets are propose: the first testing set consists of 6,375 images randomly selected from a similar distribution as the training set. The second testing set contains 15,935 images from a different distribution, representing $18^{th}$ century manuscripts. The first test evaluates baseline performance in an IID setting, while the second test assesses performance in an OOD scenario. The detail statistic is provided in section 3 of the main paper.

**Are there any errors, sources of noise, or redundancies in the dataset?** If so, please provide a description.

While the ground-truth text was double-checked by a supervisor for each annotator, we recommend additional revision of the the ground-truth texts by multiple historical document experts to minimize annotation errors.

**Is the dataset self-contained, or does it link to or otherwise rely on external resources (e.g., websites, tweets, other datasets)?** If it links to or relies on external resources, a) are there guarantees that they will exist, and remain constant, over time; b) are there official archival versions of the complete dataset (i.e., including the external resources as they existed at the time the dataset was created); c) are there any restrictions (e.g., licenses, fees) associated with any of the external resources that might apply to a future user? Please provide descriptions of all external resources and any restrictions associated with them, as well as links or other access points, as appropriate.

The dataset is entirely self-contained. It will exist, and remain constant, over time once we release it.

**Does the dataset contain data that might be considered confidential (e.g., data that is protected by legal privilege or by doctor-patient confidentiality, data that includes the content of individuals non-public communications)?** If so, please provide a description.

No.

**Does the dataset contain data that, if viewed directly, might be offensive, insulting, threatening, or might otherwise cause anxiety?** If so, please describe why.

No.

**Does the dataset relate to people?** If not, you may skip the remaining questions in this section.

No.

**Does the dataset identify any subpopulations (e.g., by age, gender)?** If so, please describe how these subpopulations are identified and provide a description of their respective distributions within the dataset.

No.

**Is it possible to identify individuals (i.e., one or more natural persons), either directly or indirectly (i.e., in combination with other data) from the dataset?** If so, please describe how.

No.

**Does the dataset contain data that might be considered sensitive in any way (e.g., data that reveals racial or ethnic origins, sexual orientations, religious beliefs, political opinions or union memberships, or locations; financial or health data; biometric or genetic data; forms of government identification, such as social security numbers; criminal history)?** If so, please provide a description.

No.

**Any other comments?** No.

### A.3 Collection Process

**How was the data associated with each instance acquired?** Was the data directly observable (e.g., raw text, movie ratings), reported by subjects (e.g., survey responses), or indirectly inferred/derived from other data (e.g., part-of-speech tags, model-based guesses for age or language)? If data was reported by subjects or indirectly inferred/derived from other data, was the data validated/verified? If so, please describe how.

The historical Ethiopic manuscripts were solely collected from Ethiopian national Archive and Library Agency (ENALA). Each instance is an image/scanned version of documents and is directly observable (see the main paper from section 3).

**What mechanisms or procedures were used to collect the data (e.g., hardware apparatus or sensor, manual human curation, software program, software API)?** How were these mechanisms or procedures validated?

After obtaining the scanned copy of the manuscript from ENALA and extracting the text-image lines, we hire individuals to annotate each text-line image. During the annotation process, all annotators have the freedom to refer to any external sources. for annotation purpose, annotation, we develop an offline tool that can be easily installed on each user's machine (see Figure 13).

**If the dataset is a sample from a larger set, what was the sampling strategy (e.g., deterministic, probabilistic with specific sampling probabilities)?**

The historical documents were collected from ENALA. While we did not have the authority to select specific documents, the workers randomly select pages, taking into account our request and the need to maintain the confidentiality of the book's information.

**Who was involved in the data collection process (e.g., students, crowdworkers, contractors) and how were they compensated (e.g., how much were crowdworkers paid)?**

the participants were students and staff members and for the raw manuscript collection and digitization we paid money as a compensation.

**Over what timeframe was the data collected? Does this timeframe match the creation timeframe of the data associated with the instances (e.g., recent crawl of old news articles)?** If not, please describe the timeframe in which the data associated with the instances was created.

The dataset was collected in March-May 2022 and the complete data creation (including preprocessing, annotation and verification were done from September 2022-February 2023.

**Were any ethical review processes conducted (e.g., by an institutional review board)?** If so, please provide a description of these review processes, including the outcomes, as well as a link or other access point to any supporting documentation.

No.

**Does the dataset relate to people?** If not, you may skip the remaining questions in this section.

No.

**Did you collect the data from the individuals in question directly, or obtain it via third parties or other sources (e.g., websites)?**

As described section 3 of the main paper, the data was collected from ENALA directly.

**Were the individuals in question notified about the data collection?** If so, please describe (or show with screenshots or other information) how notice was provided, and provide a link or other access point to, or otherwise reproduce, the exact language of the notification itself.

Yes, the scanned copies of document images were collected directly from ENALA. This request was made in person along with a letter, which also explained the objectives, goals, and the need for data in our work.

**Did the individuals in question consent to the collection and use of their data?** If so, please describe (or show with screenshots or other information) how consent was requested and provided, and provide a link or other access point to, or otherwise reproduce, the exact language to which the individuals consented.

Yes, once we met with the staff at ENALA and explained the goals of our project, they agreed to provide the data and arranged a way for delivering the documents.

**If consent was obtained, were the consenting individuals provided with a mechanism to revoke their consent in the future or for certain uses?** If so, please provide a description, as well as a link or other access point to the mechanism (if appropriate).

No.

**Has an analysis of the potential impact of the dataset and its use on data subjects (e.g., a data protection impact analysis) been conducted?** If so, please provide a description of this analysis, including the outcomes, as well as a link or other access point to any supporting documentation.

No.

### A.4 Preprocessing/cleaning/labeling

**Was any preprocessing/cleaning/labeling of the data done (e.g., discretization or bucketing, tokenization, part-of-speech tagging, SIFT feature extraction, removal of instances, processing of missing values)?** If so, please provide a description. If not, you may skip the remainder of the questions in this section.

Yes, preprocessing tasks such as image segmentation and the removal of non-Ethiopic characters were performed. Furthermore, alignments between the images and their corresponding text-line images were double-checked for each submission by the annotators and verified by a reviewer.

**Was the "raw" data saved in addition to the preprocessed/cleaned/labeled data (e.g., to support unanticipated future uses)?** If so, please provide a link or other access point to the "raw" data.

No.

**Is the software used to preprocess/clean/label the instances available?** If so, please provide a link or other access point.

Yes, here is the link for the labeling tool that we developed with the aim of fitting and making it easier for the target annotators. It is designed to accommodate their operating systems and internet service settings, allowing them to work offline when there is no internet connection. You can access the tool at this link: `https://github.com/bdu-birhanu/HHD-Ethiopic/tree/main/labeling_tool`. For preprocessing tasks, including column detection, binarization, and text-line segmentation, we utilize the OCRopus framework. You can find more information about the framework and its functionalities on their GitHub page: `https://github.com/ocropus/ocropy`

## A.5   Uses

**Has the dataset been used for any tasks already?** If so, please provide a description.

HHD-Ethiopic is a new historical handwritten Ethiopic OCR dataset for a text-line image recognition. In this work we evaluate several state-of-the-art deep learning models and an independent human-level recognition performance on a dataset, which involves comparing the performance of several human annotators with the performance of machine models. The human-level performance serves as a benchmark and in turn it also contribute to the uniqueness and quality of the dataset.

**Is there a repository that links to any or all papers or systems that use the dataset?** If so, please provide a link or other access point.

Yes, we release our dataset, code, baseline models and human-level performances at `https://github.com/bdu-birhanu/HHD-Ethiopic`.

**What (other) tasks could the dataset be used for?**

The HHD-Ethiopic dataset was specifically created to address the gap in Historical handwritten Ethiopic manuscript recognition. However, it can also be utilized to benchmark the performance of machine learning models for other scripts.

**Is there anything about the composition of the dataset or the way it was collected and preprocessed/cleaned/labeled that might impact future uses?** For example, is there anything that a future user might need to know to avoid uses that could result in unfair treatment of individuals or groups (e.g., stereotyping, quality of service issues) or other undesirable harms (e.g., financial harms, legal risks) If so, please provide a description. Is there anything a future user could do to mitigate these undesirable harms?

The datasets can be used without further considerations.

**Are there tasks for which the dataset should not be used?** If so, please provide a description.

No.

**Any other comments?**  No.

### A.6  Distribution

**Will the dataset be distributed to third parties outside of the entity (e.g., company, institution, organization) on behalf of which the dataset was created?** If so, please provide a description.

Yes, both the dataset and baseline results will be made available to the public research community for experimentation and further work on historical handwritten recognition.

**How will the dataset will be distributed (e.g., tarball on website, API, GitHub)** Does the dataset have a digital object identifier (DOI)?

The HHD-Ethiopic dataset can be downloaded from `https://github.com/bdu-birhanu/HHD-Ethiopic` or directly for the Huggingface `https://huggingface.co/datasets/OCR-Ethiopic/HHD-Ethiopic`. The images can be downloaded as a zipped file. The digital object identifie (DOI) of the dataset is: doi:10.57967/hf/0691. Our dataset has also been made public on Zenodo.org. However, we have chosen to provide it on Hugging Face and GitHub as well, as we believe these platforms are commonly used within the document image analysis and machine learning community.

**When will the dataset be distributed?**

The dataset is currently available for use in our repository.

**Will the dataset be distributed under a copyright or other intellectual property (IP) license, and/or under applicable terms of use (ToU)?** If so, please describe this license and/or ToU, and provide a link or other access point to, or otherwise reproduce, any relevant licensing terms or ToU, as well as any fees associated with these restrictions. This work is licensed under a CC-BY-4.0 International License and available at: `https://github.com/bdu-birhanu/HHD-Ethiopic` or can be directly downloaded from `https://huggingface.co/datasets/OCR-Ethiopic/HHD-Ethiopic`

**Have any third parties imposed IP-based or other restrictions on the data associated with the instances?** If so, please describe these restrictions, and provide a link or other access point to, or otherwise reproduce, any relevant licensing terms, as well as any fees associated with these restrictions.

No.

**Do any export controls or other regulatory restrictions apply to the dataset or to individual instances?** If so, please describe these restrictions, and provide a link or other access point to, or otherwise reproduce, any supporting documentation.

### A.7  Maintenance

**Who will be supporting/hosting/maintaining the dataset?**

The authors of this paper are responsible for supporting the datasets.

**How can the owner/curator/manager of the dataset be contacted (e.g., email address)?**

The curators of the dataset can be contacted via email and we provide it in the repository `https://github.com/bdu-birhanu/HHD-Ethiopic`

**Is there an erratum?** If so, please provide a link or other access point.

There is no an explicit erratum.

**Will the dataset be updated (e.g., to correct labeling errors, add new instances, delete instances)?** If so, please describe how often, by whom, and how updates will be communicated to users (e.g., mailing list, GitHub)?

Yes, we have plans to add more data to the dataset. As updates are made, we will ensure that both the documentation and our repository are updated accordingly.

**If the dataset relates to people, are there applicable limits on the retention of the data associated with the instances (e.g., were individuals in question told that their data would be retained for a fixed period of time and then deleted)?** If so, please describe these limits and explain how they will be enforced.

No.

**Will older versions of the dataset continue to be supported/hosted/maintained?** If so, please describe how. If not, please describe how its obsolescence will be communicated to users.

Any changes made to the dataset will ensure that the original version remains available, and subsequent versions, such as HHD-Ethiopic-1.1, will be released with documentation.

**If others want to extend/augment/build on/contribute to the dataset, is there a mechanism for them to do so?** If so, please provide a description. Will these contributions be validated/verified? If so, please describe how. If not, why not? Is there a process for communicating/distributing these contributions to other users? If so, please provide a description.

Yes, users can contribute to the dataset and can contact the original authors about incorporating fixes/extensions. This is encouraged. Users are free to extend or augment the dataset for their purposes.

**Any other comments?** None.

### A.8 Accessibility

1. Links to access the **dataset** and its **metadata** and **code and simulation environment**. `https://github.com/bdu-birhanu/HHD-Ethiopic`

2. **Data format**: we follow widely used data formats in OCR dataset. The actual text-line images are stored in .png format wile ground-truth texts are in .txt. the image-ground truth pair are given in .CSV formats, in addition, the images and their corresponding ground-truth are also stored in numpy format. An example of the dataset structure can be found in the README.md file of our dataset repository.

3. **Long-term preservation**: we the authors are responsible to maintain and ensure consistency of the data and it will be in our GitHub repository.

4. **Explicit license**: The dataset is licensed under a CC-BY-4.0 and the source code is under MIT license `https://github.com/bdu-bf/HHD-Ethiopic`

5. **A persistent dereferenceable identifier**: A DOI from Hugging Face, doi:10.57967/hf/0691

### A.9 Author statement

The authors have conducted a thorough review of the information presented in this document. To the best of our knowledge, the datasets included in HHD-Ethiopic are intended for research purposes and should be used in accordance with the described methodology and licenses outlined in the Accessibility section. It is important to note that the authors assume full responsibility in the event of any violation of rights.

## B Ethiopic writing systems

Ethiopic script is an ancient writing system used primarily in Ethiopia and Eritrea. With its origins dating back to the $4^{th}$ century AD [13]. The script is characterised by its unique syllabic structure, which combines consonants and vowels to form complex characters. In literature the Ethiopic writing system also named with various names including "Abugida", "Amharic", "Ge'ez", and "Fidel".

Ethiopic script has been a significant cultural and linguistic heritage of the region, playing a vital role in preserving the rich history and traditions of Ethiopia. It is primarily used for writing over 27 languages including the Amharic and Tigrinya languages, among others. As depicted in Figure 7, the script has a distinct visual appearance, characterized by its curved and geometric shapes, making it visually distinctive and is written and read, as English, from left to right and top to down [5].

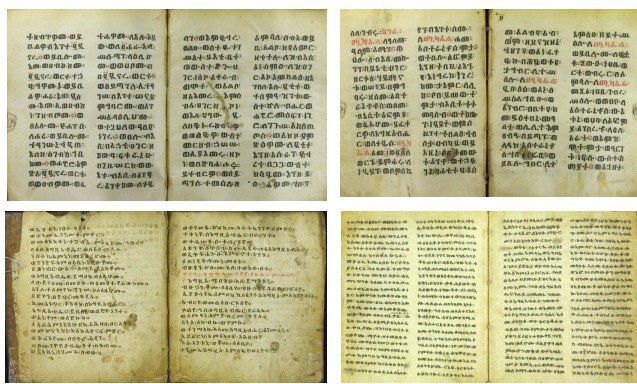

Figure 7: Sample historical handwritten Ethiopic manuscripts

Despite the long history of the Ethiopic script, it has encountered numerous challenges in the digital world due to its low-resource nature [7, 16]. Issues such as limited digitized fonts, linguistic tools, and datasets have posed obstacles in the fields of natural language processing and document image analysis technologies.

The Ethiopic script poses unique challenges for machine learning due to the scarcity of available resources. This script is characterized by its complex orthographic identities and visually similar characters. Comprising over 317 distinct characters, including approximately 280 characters organized in a 2D matrix format known as Fidel-Gebeta (Figure 8), along with 20 digits and 8 punctuation marks (Figure 9).

As depicted in Figure 8, the Ethiopic script consists of 34 consonant characters, which serve as the base for deriving additional characters using diacritics. These diacritics can be found as small marks placed on the top, bottom, left, or right sides of the base character. Furthermore, specific vowel characters are formed by shortening either the left or right leg of consonant characters, as demonstrated in columns 4 (shortening left leg) and 7 (shortening right leg) of the fidel-Gebeta. The vowels, derived from these consonants, span from 1 to 12 and correspond to the respective columns.

For example, in the second row of the fidel-Gebeta, the consonant character ለ represents the sound "le" in Ethiopic. From this base character, various vowel characters emerge, such as:

- ሉ is formed by adding a horizontal diacritic at the middle left side of the base character and represents the sound "lu".
- ሊ is formed by adding a horizontal diacritic at the bottom left leg of the base character and represents the sound "li".
- ላ is formed by shortening the left leg of the base character and represents the sound "la".

These examples showcase the versatility of the Ethiopic script, where modifying the diacritics or leg lengths of consonant characters allows for the representation of different vowel sounds.

|    |     | 1 | 2 | 3 | 4 | 5 | 6 | 7 | 8 | 9 | 10 | 11 | 12 |
|----|-----|---|---|---|---|---|---|---|---|---|----|----|----|
|    |     | *ä/e* | *u* | *i* | *a* | *ē* | *ə* | *o* | *ʷä/ue* | *ʷi/u* | *ʷa/ua* | *ʷē/uē* | *ʷə* |
| 1  | h   | ሀ | ሁ | ሂ | ሃ | ሄ | ህ | ሆ |   |   |   |   |   |
| 2  | l   | ለ | ሉ | ሊ | ላ | ሌ | ል | ሎ |   |   | ሏ |   |   |
| 3  | ḥ   | ሐ | ሑ | ሒ | ሓ | ሔ | ሕ | ሖ |   |   | ሗ |   |   |
| 4  | m   | መ | ሙ | ሚ | ማ | ሜ | ም | ሞ |   |   | ሟ |   |   |
| 5  | ś   | ሠ | ሡ | ሢ | ሣ | ሤ | ሥ | ሦ |   |   | ሧ |   |   |
| 6  | r   | ረ | ሩ | ሪ | ራ | ሬ | ር | ሮ |   |   | ሯ |   |   |
| 7  | s   | ሰ | ሱ | ሲ | ሳ | ሴ | ስ | ሶ |   |   | ሷ |   |   |
| 8  | š   | ሸ | ሹ | ሺ | ሻ | ሼ | ሽ | ሾ |   |   | ሿ |   |   |
| 9  | q   | ቀ | ቁ | ቂ | ቃ | ቄ | ቅ | ቆ | ቈ | ቊ | ቋ | ቌ | ቍ |
| 10 | b   | በ | ቡ | ቢ | ባ | ቤ | ብ | ቦ |   |   | ቧ |   |   |
| 11 | v   | ቨ | ቩ | ቪ | ቫ | ቬ | ቭ | ቮ |   |   | ቯ |   |   |
| 12 | t   | ተ | ቱ | ቲ | ታ | ቴ | ት | ቶ |   |   | ቷ |   |   |
| 13 | č   | ቸ | ቹ | ቺ | ቻ | ቼ | ች | ቾ |   |   | ቿ |   |   |
| 14 | ḫ   | ኀ | ኁ | ኂ | ኃ | ኄ | ኅ | ኆ | ኈ | ኊ | ኋ | ኌ | ኍ |
| 15 | n   | ነ | ኑ | ኒ | ና | ኔ | ን | ኖ |   |   | ኗ |   |   |
| 16 | ñ   | ኘ | ኙ | ኚ | ኛ | ኜ | ኝ | ኞ |   |   | ኟ |   |   |
| 17 | '   | አ | ኡ | ኢ | ኣ | ኤ | እ | ኦ |   |   | ኧ |   |   |
| 18 | k   | ከ | ኩ | ኪ | ካ | ኬ | ክ | ኮ | ኰ | ኲ | ኳ | ኴ | ኵ |
| 19 | x   | ኸ | ኹ | ኺ | ኻ | ኼ | ኽ | ኾ | ዀ | ዂ | ዃ | ዄ | ዅ |
| 20 | w   | ወ | ዉ | ዊ | ዋ | ዌ | ው | ዎ |   |   |   |   |   |
| 21 | '   | ዐ | ዑ | ዒ | ዓ | ዔ | ዕ | ዖ |   |   |   |   |   |
| 22 | z   | ዘ | ዙ | ዚ | ዛ | ዜ | ዝ | ዞ |   |   | ዟ |   |   |
| 23 | ž   | ዠ | ዡ | ዢ | ዣ | ዤ | ዥ | ዦ |   |   | ዧ |   |   |
| 24 | y   | የ | ዩ | ዪ | ያ | ዬ | ይ | ዮ |   |   |   |   |   |
| 25 | d   | ደ | ዱ | ዲ | ዳ | ዴ | ድ | ዶ |   |   | ዷ |   |   |
| 26 | ǧ   | ጀ | ጁ | ጂ | ጃ | ጄ | ጅ | ጆ |   |   | ጇ |   |   |
| 27 | g   | ገ | ጉ | ጊ | ጋ | ጌ | ግ | ጎ | ጐ | ጒ | ጓ | ጔ | ጕ |
| 28 | ṭ   | ጠ | ጡ | ጢ | ጣ | ጤ | ጥ | ጦ |   |   | ጧ |   |   |
| 29 | č̣  | ጨ | ጩ | ጪ | ጫ | ጬ | ጭ | ጮ |   |   | ጯ |   |   |
| 30 | p   | ጰ | ጱ | ጲ | ጳ | ጴ | ጵ | ጶ |   |   | ጷ |   |   |
| 31 | ṣ   | ጸ | ጹ | ጺ | ጻ | ጼ | ጽ | ጾ |   |   | ጿ |   |   |
| 32 | ṣ́  | ፀ | ፁ | ፂ | ፃ | ፄ | ፅ | ፆ |   |   |   |   |   |
| 33 | f   | ፈ | ፉ | ፊ | ፋ | ፌ | ፍ | ፎ |   |   | ፏ |   |   |
| 34 | **p** | ፐ | ፑ | ፒ | ፓ | ፔ | ፕ | ፖ |   |   | ፗ |   |   |

Figure 8: Fidel-Gebeta: the row-column matrix structure of Ethiopic characters. The first column shows the consonants, while the following columns (1-12) illustrate syllabic variations (obtained by adding diacritics or modifying parts of the consonant).

Ethiopic numerals also called Ge'ez numerals, are a numeric system traditionally used in Ethiopic writing. These numeral system has its own distinct symbols for representing numbers, which are different from the Arabic or Roman numerals commonly used in many other parts of the world. The system has a base of 10, with unique characters for each digit from 1 to 9, as well as special symbols for tens, hundreds, and thousands (Figure 9). For example:

- Ethiopic symbol ፩ is similar to the Arabic numeral 1.
- symbol ፵፬ is similar to the Arabic numeral 44.
- symbol ፱፼፱፻፺፱ similar to the Arabic numeral 99999.
- symbol ፼፪ is similar to the Arabic numeral 10002.
- symbol ፲፪፻፴፫ similar to the Arabic numeral 1233.

Though modern Arabic numerals dominate daily life and official documents, understanding Ethiopic numerals is vital for deciphering historical texts and preserving cultural heritage.

In the Ethiopic writing system, punctuation marks convey meaning and guide text interpretation (see Figure 9). Understanding their usage is vital for clear and effective written communication in Ethiopic script.

| C | ፪ | ፫ | ፬ | ፭ | ፮ | ፯ | ፰ | ፱ | ፲ |
|---|---|---|---|---|---|---|---|---|---|
| 1 | 2 | 3 | 4 | 5 | 6 | 7 | 8 | 9 | 10 |
| ፳ | ፴ | ፵ | ፶ | ፷ | ፸ | ፹ | ፺ | ፻ | ፼ |
| 20 | 30 | 40 | 50 | 60 | 70 | 80 | 90 | 100 | 10000 |

**a**

| ※ | ፡ | ። | ፣ | ፤ | ፥ | ፧ | ፨ |
|---|---|---|---|---|---|---|---|
| section mark | word separator | full stop (period) | comma | semicolon | colon | question mark | paragraph separator |

**b**

Figure 9: Numbering system (a) and punctuation marks (b) in Ethiopic script

The complexities of symbols within the Ethiopic script present significant challenges for machine learning tasks, requiring attentive approaches to achieve accurate recognition and analysis. An example of these challenges is the non-standardized usage of punctuation marks 10 and variations in writing styles, as depicted in Figure 7. These factors contribute to the difficulties encountered in the development of Ethiopic OCR systems.

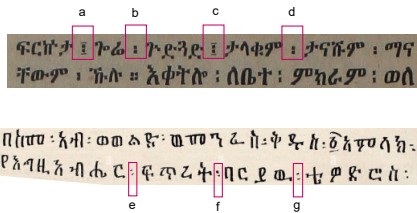

Figure 10: Examples of punctuation usage and writing Styles: As shown by the red rectangle and labeled by [a, b, c, d], there is typically a space before and after the punctuation mark. In contrast, the punctuation marks labeled by [e, f, g] do not have any space before or after them. The punctuation marks labeled by a and c serve as list separators and are distinct from the other punctuation marks, which are used as word separators.

## C  Methods and implementation details

In this section, we provide additional details of models implemented and evaluated on our HHD-Ethiopic OCR dataset. We evaluate several state-of-the-art methods, which can be broadly grouped as CTC-based, Attention, and Transformer-based. However, our primary focus in this section is on the CTC-based model, which is designed to operate effectively in lower resource settings. This is because the other CTC, attention and Transformer-based model ( evaluated on this new datasets) are validated from previous works [9, 10, 14, 17, 18] and involves extensive hyperparameters, making it more suited for higher-resource environments. These SOTA methods are implemented using the open-source toolbox, mmocr: `https://github.com/open-mmlab/mmocr`.

### C.1  Baseline models

The implementation of the CTC-based model follows a typical pipeline depicted in Figure 11. In case of Plain-CTC, initially, the preprocessed images are passed through a convolutional neural network (CNN) backbone, which extracts relevant image features using a series of convolutional and pooling layers.

The output features from the CNN backbone are reshaped and subsequently fed into a a Long Short-term Memory (LSTM) network with connectionist temporal classification (CTC) network. This combination enables the model to effectively capture the temporal dependencies between the image features and the corresponding text labels. The RNN layer incorporates two Bi-directional LSTM units to learn sequential patterns and generate a $[(c+1) \times T]$ matrix of Softmax probabilities for

each character at each time-step, where c and T denote the number of characters and the length of maximum time-step. Finally, a the CTC converts the intermediate representations into the final output text predictions.

The alternative CTC-based approach, referred to as Attn-CTC within this paper and previously introduced for Amharic text recognition[6], extends the Plain-CTC methodology by incorporating an attention mechanism into the CTC layers. The rationale behind incorporating the attention layer lies in leveraging its capacity to derive a more potent hidden representation through a weighted contextual vector. This model comprises a combination of CNN and LSTM as the encoding module. The output of this module feeds into the attention module, and subsequently, the decoded output string is obtained through the CTC layer.

During training, the CTC algorithm calculates the likelihood of the output sequence given the input sequence and uses it as the objective function [12, 15]. The training process maximizes this likelihood, which, in turn, maximizes the probability of the correct output sequence. The loss that is minimized during training is the negative of this likelihood, which can be defined as:

$$CTC_{loss} = -log \sum_{(y,x) \in S} p(y/x) \tag{1}$$

where x and y denote pair of input and output sequences in sample dataset S respectively and the probability of label sequence for a single pair p(y/x) is computed by multiplying the probability of labels along a specific path $\pi$ for the overall time steps T and it can be defined as:

$$P(y/x) = \prod_{t=1}^{T} p(a_t, \pi) \tag{2}$$

where a is a character in the specified path and p(a) is its probability on each time-step on that path.

Once training and evaluating the OCR model with network settings proposed in [4, 6], we employed Bayesian optimization for the selection of hyperparameters, with the CTC validation loss serving as the criteria for optimization. Bayesian optimization captures the relationship between the hyperparameters and the CTC validation loss, iteratively updating and refining the model as it explores different hyperparameter configurations ( see ref [3]for details) that yields lower CTC validation loss values. This approach allowed us to effectively tune our model and enhance its performance, contributing to the overall success of our text-image recognition model.

The source code for hyperparameter selection and training procedures are provided at `https://github.com/bdu-bf/HHD-Ethiopic`.

The recognition performance of all human-level and baseline models evaluated in this work is reported using the character error rate (CER) and Normalized Edit Distance (NED) metrics. All results reported with these two metrics are converted to 100%. The CER metric can be computed as follows,

$$CER(T, P) = \left( \frac{1}{c} \sum_{m \in T, n \in P} ED(m, n) \right) \times 100, \tag{3}$$

where $c$ denotes the total number of characters in the ground-truth, $t$ and $p$ denote the ground-truth labels and predicted respectively, and $ED(m, n)$ is the Levenshtein edit-distance between sequences $m$ and $n$.

while the NED metric is computed as:

$$NED = \left( \frac{1}{N} \sum_{i=1}^{N} \frac{ED(m_i, n_i)}{\max(l_i, \hat{l}_i)} \right) \times 100 \tag{4}$$

---

[6] https://deephyper.readthedocs.io/en/latest/index.html

where N is the maximum number of paired ground truth and prediction strings, *ED* is the Levenshtein edit distance, $m_i$ and $n_i$ denote the predicted text and the corresponding ground truth (GT) string, respectively, and $l_i$ and $\hat{l}_i$ are their respective text lengths.

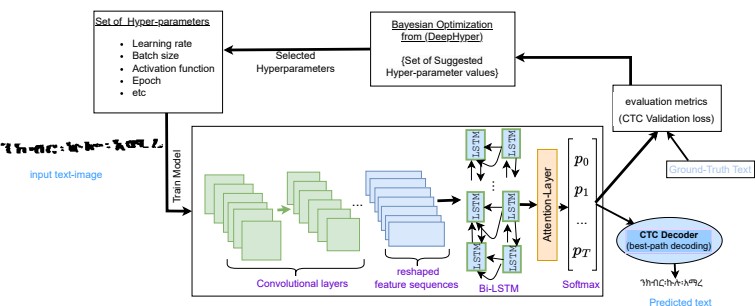

Figure 11: A typical view of the proposed model and set of best hyper-parameters value selection using Bayesian optimization from DeepHyper[6]. The output denoted by $p_0, p_1...p_T$, is a matrix of Softmax probabilities with dimensions $[(c+1) \times T]$, where c is the number of unique characters in the ground-truth text and T is the length of the input time-step to the LSTM layers. The validation loss was utilized as the metric for tuning the hyperparameters. To obtain the final output sequence from the predicted probabilities produced by the model, we use the best-path decoding strategy.

## C.2 Training details and configurations

During our experiments, we employed various hyperparameter settings, including those selected by Bayesian Optimization [3] specifically for the CTC-based models. Training and evaluation were performed on a single NVIDIA RTX A6000 GPU for all the baseline models. Except for the TrOCR transformer-based models, the training process for each individual model required a wall time of less than 2.5 hours. However, when considering that there were 10 experiments conducted, with each model trained 10 times, the cumulative wall training time is going to be 25 hours each (i.e 10 experiments * 10 runs * 2.5 = 250 hours in total). Additional details regarding the training can be found in the provided at `https://github.com/bdu-bf/HHD-Ethiopic`.

For the CTC-based baseline models, we trained them multiple times with different hyperparameter values, including epochs ranging from 10 to 100, employing a trial-and-error approach and utilizing the hyperparameters suggested by Bayesian Optimization. In this paper, we report the results obtained from the two CTC-based models (without attention) achieving better CER in 15 epochs. Additionally, the attention-CTC models showed improved performance as we trained them for more epochs. The reported results, for attention-CTC models, in the main paper were trained for 100 epochs.

Despite the TrOCR [14] model has been reported to achieve state-of-the-art performance in the original paper, it has a significant drawback due to its large number of parameters. Our attempts to fine-tune the TrOCR model using our HHD-Ethiopic dataset, following the provided tutorial, faced substantial computational challenges. Training the model for just 3 epochs on a single NVIDIA RTX A600 GPU took over 24 hours, resulting in comparatively lower performance compared to the CTC-based baseline models. Considering our focus on low-resource settings, we prioritize optimizing our time and resources effectively. Hence, as it is not suitable for training in resource-constrained environments, we do not recommend utilizing the TrOCR model for Ethiopic text recognition. Instead, we prioritize exploring alternative models ( such as the smaller CTC-based methods discussed in the main paper) which balance between computational efficiency and performance to ensure the feasibility of the OCR system in limited resources. However, if you possess significant computing resources, using synthetic data and conducting more extensive training iterations on those models could lead to an improvement in recognition performance for historical handwritten Ethiopic manuscripts.

We also evaluated various other models [9, 10, 17, 18] using our HHD-Ethiopic dataset. Although these models still have a relatively high number of parameters in comparison to the CTC-based

models ( the plain and Attn-CTC), they remain more manageable in low-resource settings. Despite the increased parameter count, we run these models for 25 epochs using limited computational resources. We achieved an improved recognition performance compared to the results presented in the TrOCR paper. By balancing performance and resource demands, the models [9, 10, 17, 18] present a viable option for practical deployment and utilization, especially in situations where computational resources are constrained.

Due to the limited number of experimental runs conducted for [9, 10, 14, 17, 18] baseline models, we decided not to include box plots for all baseline models in the main paper. Box plots are commonly used to visualize results distribution across multiple runs, allowing for the assessment of variations and identification of outliers. Since a box plot is not suitable for representing a single experiment, we have illustrated the learning curve of the four models (ABINet, ASTER, SVTR and CRNN) in Figure 12. This learning curve illustrates the recognition performance on both IID and OOD test sets using the CER and metric across 25 epochs. For detailed configurations of each baseline OCR model and the implementation of Bayesian optimization, please refer to our GitHub repository at the following link at `https://github.com/bdu-bf/HHD-Ethiopic`.

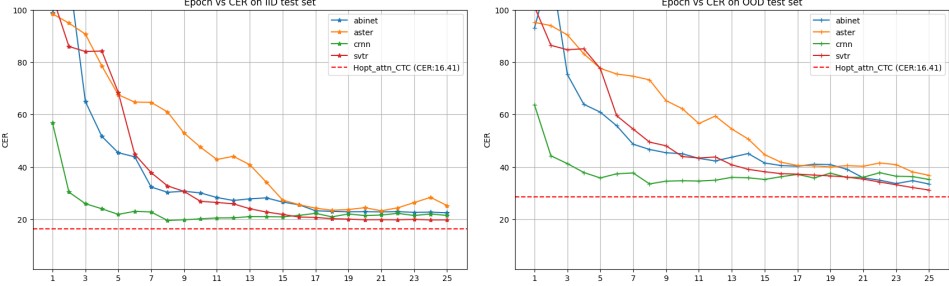

Figure 12: Learning curve on IID and OOD test data. CER[1] on IID test set (left), CER on OOD test set (right) across 25 epochs for ASTER, ABInet, SVTR, and CRNN models. In all plots, the red horizontal line represents the CER value of the Hopt-attn-CTC network on IID and OOD data respectively.

Based on learning curve depicted in Figure 12, we can conclude that all models would perform better as we train for longer epochs. Within the first 25 epochs, SVTR outperforms the others, while ASTER is the least performer. We are limited to running for 25 epochs due to time and computational resources. The red horizontal line in both the right and left plots represents the CER for Hopt-attn-CTC model. This line serves as our benchmark, as it represents the best-performing model.

### C.3 Data collection and annotation process

The Ethiopic script, one of the oldest in the world, is underrepresented in the fields of document image analysis (DIA) and natural language processing (NLP). This is due to the lack of attention from researchers in these fields and the absence of annotated datasets suitable for machine learning. However, in recent times, there has been a significant increase in interest from individuals involved in computing and digital humanities. As part of this growing attention, we have contributed by preparing this first sizable historical handwritten dataset for Ethiopic text-image recognition. The primary source of these documents is the Ethiopian National Archive and Library Agency (ENALA), spanning from the $18^{th}$ to the $20^{th}$ century. To ensure privacy, each page is randomly sampled from about seven different books covering cultural and religious related contents. After obtaining scanned copies of

---

[1] Please note that the CER can exceed 100% when the predicted text is much longer than the ground truth. Excessive length leads to an edit distance surpassing the ground truth's character count. For instance, if the ground truth is 'ab' and the prediction is 'abced' the edit distance is 3 compared to the ground truth's 2 characters. This results in a ratio of 1.5*100=150 (see equations 3). In contrast, NED ranges from 0 to 100%, where values close to 0 are better, while values closer to 100% are indicative of poorer performances in both metrics.

the documents from ENALA, we utilize the OCRopus[2] OCR framework and the ground-truth text annotation process is described as follows:

The annotation process can be grouped in three phase:

- **Phase-I**: In this phase, we hired 14 individuals who are familiar with the Ethiopic script. Out of the 14, 12 were assigned the task of annotation, while the remaining two served as supervisors responsible for follow-up the annotation process and ensuring the completeness of each annotation submission. Additionally, the supervisors were responsible for multiple tasks, including monitoring the progress of each annotator, providing assistance when issues arose, making decisions to address any problems encountered during the annotation process, checking alignment consistency between images and ground-truth at each phase of the annotator's submission, and making necessary corrections in case of errors. Throughout the annotation process, all annotators and supervisors had the freedom to refer to any necessary references.

- **Phase-II**: Once we have all the annotated text-line images from phase-I, we divide the text-image into training and test sets. For the training set, we reserve all text line images from the $19^{th}$ and $20^{th}$ centuries, as well as a few documents with unknown publication dates. The test set is exclusively composed of text line images from the $18^{th}$ century. Additionally, we randomly sample another test set, which constitutes 10% of the training set. We call this randomly selected set as **Test-set-I**, which allows us to evaluate the baseline performance in the classical IID (Independently and Identically Distributed) setting.

  On the other hand, the test set that is drawn from a different distribution than the training set, known as Out-Of-Distribution (OOD), is called **Test-set-II**. This setup enables us to assess the performance in real scenarios where the test set differs from the training distribution.

- **Phase-III**: In this phase, we hired approximately 20 individuals who are familiar with the Ethiopic script, along with one historical expert for the second round of annotation and request them to submit within 5 weeks. This annotation phase has the following two objectives:

  - to ensure the quality of the test set.
  - to evaluate the human-level performance in historical Ethiopic script recognition, which serves as a baseline for comparison with machine learning models.

  Out of the 20 individuals hired, only 13 annotators successfully completed the annotation task within the specified submission deadline, while the remaining individuals failed and resigned from the task. Among the 13 successful annotators, the first group comprised 9 people who transcribed text-line images from the first test set, which consisted of 6,375 randomly selected images from the training set. The second group consisted of 4 people who transcribed the second test set, consisting of 15,935 images from the $18^{th}$ century.

  With the exception of the expert reviewer, who was allowed to use external references, all annotators in this phase were instructed to perform the task without the use of references. Detailed data from each annotator was documented as metadata for future reference and can be accessed from our GitHub repository. One observation we made during this annotation process was that some annotators anonymously shared information, despite our efforts to ensure data confidentiality. However, despite this limitation, we have successfully compute the human-level performance for each annotator and have reported the results accordingly.

Considering the resources available to the annotators, including computing infrastructure and internet access, we developed a simple user-friendly tool with a easy to use Graphical User Interface (GUI) for the annotation process. The tool is depicted in Figure 13.

Each annotator's machine was equipped with this tool, enabling them to work offline when internet access was unavailable. Additionally, we provided them with a comprehensive *README* file and instructed them on how to utilize the annotation tool.

---

[2]https://github.com/ocropus/ocropy

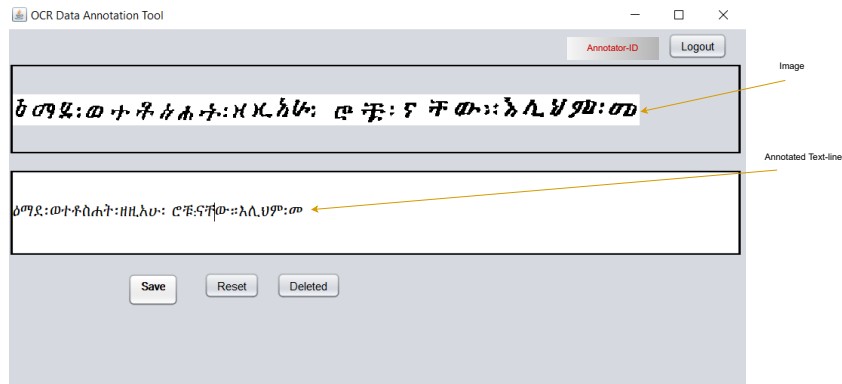

Figure 13: Text-line image annotation tool

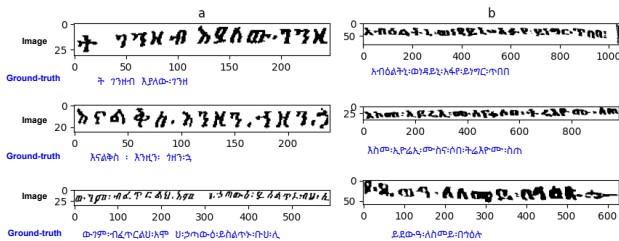

Figure 14: Sample text-line images and ground-truth for HHD-Ethiopic: a) Training-set. b) Test-set.

## C.4    Dataset statistical overview and comparisons

This section provides a detailed description of the characteristics of the HHD-Ethiopic dataset. These characteristics include the diversity of content, variations in image quality, distribution of image sizes in the trainin and test sets, the number of samples per class, and a comparison with related datasets.

Examples of sample page images are illustrated in Figure 15, showcasing pages from various publication years (categorized as $18^{th}, 19^{th}, 20^{th}$, and unknown date of publication). In addition, Figure 16 displays page images categorized by image quality, which ranges from bad to medium and good. It's important to note that documents of insufficient quality, falling below the "bad" threshold, are excluded during the process of text line extraction.

The histogram in Figure 18 illustrates the distribution of text-line image sizes (width and height) across the training set and two test sets. Additionally, access to the distribution of characters for each class (i.e., the frequency of characters within the 306 unique characters) in both the training and test sets is available at `https://github.com/bdu-birhanu/HHD-Ethiopic/tree/main/Dataset/` `distribution_of_characters`.

To better represent characters that are infrequent or absent in the training set, we have employed a solution involving the generation of synthetic images. Each character is incorporated into synthetic images approximately 200 times on average. In our scenario, we have identified characters that occur 20 times or less. About 1200 newly generated synthetic text-line images featuring these underrepresented characters are provided on Hugging Face:`https://huggingface.co/datasets/OCR-Ethiopic/` `HHD-Ethiopic/tree/main/train/train_raw/under_represented_char_synth`. Figure 17 depicts these characters along with their corresponding frequencies in the training set.

Though it may not be fair to directly compare datasets from distinct settings, we provide a comparisons between our historical handwritten (HHD-Ethiopic) dataset and the existing collections of modern printed, modern handwritten, and scene text datasets for the task of Ethiopic script recognition. The summary of comparisons is is given in Table C.4.

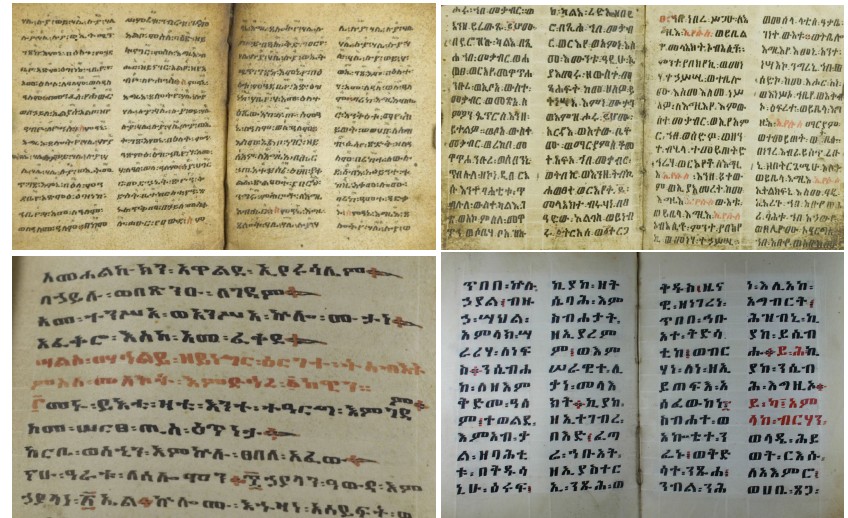

Figure 15: Sample page images ranging from $18^{th}$, $19^{th}$, $20^{th}$ centuries, as well as images of unknown publication dates, arranged from top left, top right, bottom left and bottom right respectively.

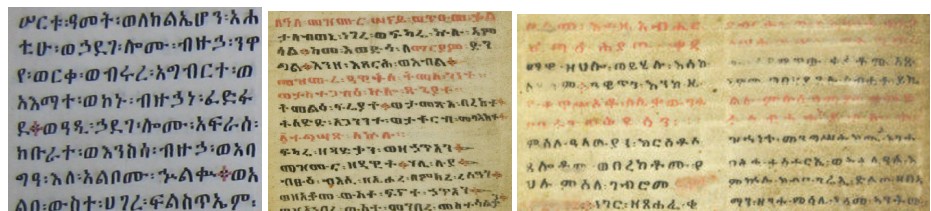

Figure 16: Sample page image images with good(left) , medium(middle) and bad (right) quality.

Table 4: Summary of publicly available datasets for Ethiopic script

| Dataset-type | image-type | # images | # uniq-chars | # test-sample | annotations |
|---|---|---|---|---|---|
| Printed[5] | real | 40,929 | 280 | 2,907 | line-level |
| | synthetic | 296,408 | 280 | 15724 | line-level |
| Scene[8] | real | 15,39 | 302 | 9,257 | word-level |
| | synthetic | 2.8M | 302 | - | word-level |
| Handwritten[1] | real/modern | 12,064 | 300 | 1,2064 | word-level |
| | Augmented | 33,672 | - | * | word-level |
| Handwritten[2] | real/modern | 10,932 | 265 | - | word-level |
| **Our** (HHD-Ethiopic) | real/historical | 79,684 | 306 | 22,310 | line-level |
| | synthetic | 100,000 | 306 | * | line-level |

- denotes information that is unavailable/ not given
* denotes data that has not been utilized for testing

## C.5 Sample predicted texts

Sample images with the corresponding ground truth, model prediction and the edit distance between the ground truth and the prediction at line level is shown in Figure19

In text lines where characters with low occurrence rates appear in the ground truth of the training set often leads to an increased edit distance between the ground truth and the predicted texts during test time. This pattern is demonstrated by sample examples depicted in Figure.20

| Char | Freq | Char | Freq | Char | Freq | Char | Freq | Char | Freq | Char | Freq | Char | Freq | Char | Freq | Char | Freq |
|---|---|---|---|---|---|---|---|---|---|---|---|---|---|---|---|---|---|
| ድ | 20 | ማ | 14 | ጥ | 7 | ⷈ | 5 | ፦ | 4 | ኸ | 3 | ኪ | 1 | ኽ | 1 |
| ፏ | 18 | ኣ | 14 | ጮ | 7 | ⷍ | 5 | ፔ | 4 | ቸ | 3 | ዥ | 1 | ⷎ | 1 |
| ፣ | 17 | ዧ | 13 | ፍ | 6 | ⷍ | 5 | ፨ | 4 | ኧ | 2 | ፕ | 1 | ፔ | 0 |
| ፄ | 17 | ፐ | 12 | ፉ | 6 | ⷓ | 5 | ቢ | 4 | ዣ | 2 | ኁ | 1 | ፄ | 0 |
| ኑ | 17 | ፐ | 12 | ⷐ | 6 | ፄ | 5 | ቪ | 3 | ⷔ | 2 | ኚ | 1 | | |
| ቡ | 17 | ⷐ | 10 | ፝ | 6 | ኽ | 5 | ፐ | 3 | ፐ | 2 | ኽ | 1 | | |
| ኮ | 16 | ዩ | 9 | ⷕ | 6 | ⷐ | 4 | ⷐ | 3 | ፣ | 1 | ⷖ | 1 | | |
| ፅ | 16 | ፐ | 9 | ኵ | 6 | ⷐ | 4 | ቪ | 3 | ዩ | 1 | ፤ | 1 | | |
| ⷐ | 15 | ኑ | 8 | ፍ | 6 | ፄ | 4 | ፍ | 3 | ፄ | 1 | ፤ | 1 | | |
| ኧ | 14 | ⷐ | 7 | ⷐ | 6 | ፏ | 4 | ⷐ | 3 | ፍ | 1 | ፍ | 1 | | |

Figure 17: Frequency distribution of underrepresented characters occurring 20 times or less in the training set. zero in the frequency column refers to the characters that exit in the test set but not in the training set.

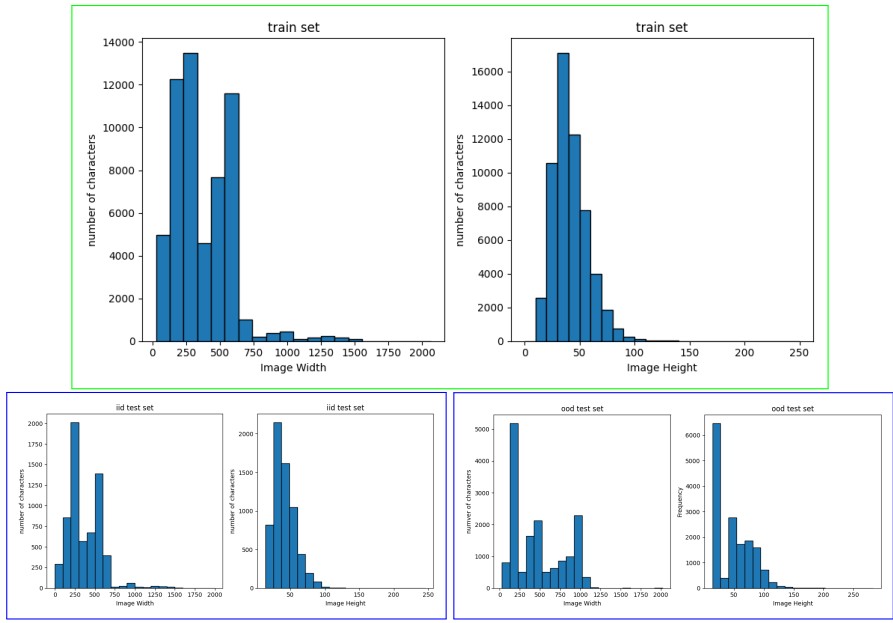

Figure 18: A histogram for distribution of image sizes in the HHD-Ethiopic dataset: a) Training-set (top). b) IID test-set (bottom left), c) OOD test set(bottom right).

Finally, we (the authors) believe that this supplementary material serves as an invaluable resource for reproducing the reported results and conducting further research on historical Ethiopic OCR. It encompasses crucial contents, including detailed information on the dataset and its preparation, strategies employed for training the baseline model, and additional essential information required for replicating the findings. This supplementary material also includes access links to the dataset and source code, enabling researchers to easily access and utilize these resources. By making use of this comprehensive supplementary material, researchers can gain deep insights into the HHD-Ethiopic dataset, the training process of the baseline OCR model, and other necessary details for accurately reproducing the results and to use this new OCR dataset. This comprehensive resource significantly supports individuals interested in working on Ethiopic OCR, providing a benchmark for their machine learning models and contributing to the advancement of research in these field.

GT Text: እስመ፡ኢጸሬኢ፡ሙስና፡ሰበ፡ትሬእየሙ፡ለጠ
Pred Text: እከሙ፡ኢይሬኢ፡መስና፡ሰበ፡ትረእየሙ፡ሰመ
Edit Distance: 9

GT Text: ቢበን፡ይመውቱ፡ወከማሁ፡ይትጋኑሉ፡አብዳ
Pred Text: ቢዋን፡ይመውቱ፡ወከማሁ፡ይትጋኑሉ፡አብጿ
Edit Distance: 4

GT Text: ጉ፡እለ፡አልበሙ፡ልብ፡ወየጋድን፡ለባዕድ፡ብዕ
Pred Text: ጉ፡እለ፡አልበሙ፡ልብ፡ወየጋቡን፡ዘግዕድ፡ብስ
Edit Distance: 6

GT Text: ወባሕቱ፡
Pred Text: ወባሕቱ
Edit Distance: 2

GT Text: ወይሰመይ፡አስማቲሆሙ፡በበ፡በውርቲሆሙ
Pred Text: ወይሰመይ፡ሕስማቲሆሙ፡በ፡በሐውርቲሆሙ፡
Edit Distance: 7

Figure 19: Sample text-line images with their corresponding ground-truth and prediction texts

GT Text: እስከ፡ማዕዚኑ፡ትኔንኑ፡ዓመፃ፡
Pred Text: እስከ፡ማዕዚኑ፡ትፍገኑ፡ዓመዊ
Edit Distance: 5

GT Text: መጸ፡እባክ፡ሃይማኖት
Pred Text: መጺኡ፡እንከ፡ሃይመኖት
Edit Distance: 6

GT Text: ዢሞ፡መዓጅሞ፡መዓጅ
Pred Text: ገርም፡መዓድ፡ም፡መዓድ
Edit Distance: 6

GT Text: ጶ፡ፓፒሮስ፡ኢየአክሮ፡
Pred Text: ክ፡ምጥሮስ፡ኢየአክሮስ
Edit Distance: 7

Figure 20: Examples of prediction errors for underrepresented characters. The characters marked in red within the ground-truth text are less frequent characters and are wrongly predicted.