# OpenReview forum: "HHD-Ethiopic: A Historical Handwritten Dataset for Ethiopic OCR with Baseline Models and Human-level Performance"
_NeurIPS.cc/2023/Track/Datasets_and_Benchmarks — Submitted to NeurIPS 2023 Datasets and Benchmarks_

### Official Review · Reviewer_BcAH · 2023-07-20
**More methods can be compared on the proposed dataset**

**Rating:** 7
**Confidence:** 4
**Clarity:** This paper is well-written.

**Strengths:**

1. This paper is well-written and well-organized.
2. Detailed analyses are provided for the experimental results.
3. The proposed dataset is helpful for the research on Ethiopic OCR.

**Additional Feedback:**

See comments in Opportunities For Improvement.

**Correctness:**

The datasets are constructed in a sound way; the experiment design is appropriate and performed correctly.

**Documentation:**

There is sufficient detail on data collection and organization, availability and maintenance, and ethical and responsible use.

**Limitations:**

No potential negative societal impact.

**Opportunities For Improvement:**

1. Text line recognition accuracy is a commonly-used evaluation metric in the field of text recognition, and the authors should add the corresponding results.
2. In Latin text recognition tasks, synthetic datasets are usually used for training. In Ethiopic OCR, can using synthetic datasets for training effectively improve the model's performance?
3. More methods [1,2] that are originally proposed for Latin text recognition need to be listed as baseline models.
[1] Du, Yongkun, et al. "SVTR: Scene Text Recognition with a Single Visual Model." IJCAI, 2022.
[2] Bautista, Darwin, and Rowel Atienza. "Scene text recognition with permuted autoregressive sequence models." ECCV, 2022.
4. It would be better to use vector graphics or high-resolution images.

**Relation To Prior Work:**

Yes.

**Summary And Contributions:**

This paper introduces a new OCR dataset for historical handwritten Ethiopic text recognition. The authors evaluate the human-level performance to provide a baseline for comparison with learning-based models. In addition, the authors evaluate several SOTA Transformer-based and CTC-based methods.

---

> ### Author Response · Authors · 2023-08-19
>
> # Response to reviewer [BcAH]
> Thank you for reviewing our work. Your input has helped us improve the paper. We've addressed your questions and comments below.
>
> [**evaluation metrics**] we agree that text-line accuracy can be used as a metric for text-image recognition mostly having short sequence texts such as in scene text recognition where input often comprises short text like word images. However, our dataset consists of extensive text sequences, some spanning up to 46 characters. This considerable length renders the text-line recognition accuracy metric excessively challenging to evaluate fairly.
>
> Our main focus is on reflecting the effort required by a human editor to correct OCR-generated text. To illustrate, even a single character omission within a lengthy text line, for example, 45 characters in our context; results in a complete loss of text line accuracy. This scenario, which is atypical in the long handwritten text recognition literature. However, we provide additional metrics, NED,  that is recommended in this case and also widely used in text recognition competitions ( e.g https://arxiv.org/pdf/1909.07741.pdf)  in the revised version (see table 2 and table 3 of the revised main paper).We have also provide additional  examples, such as the edit distance between ground truth and predicted text lines, in the revised supplementary material (Appendix, section C.5).
>
> [**Synthetic text-line image**] we agree that using large amounts of synthetic data for training can improve performance. But our goal was to create a  text-line level historical handwritten dataset considering a low-resource computing  environment. We generated synthetic text-line images especially for underrepresented characters in the dataset and added these samples into the HDD-Ethiopic dataset directory in Hugging Face: https://huggingface.co/datasets/OCR-Ethiopic/HHD-Ethiopic/tree/main/train/train_raw/under_represented_char_synth. We will keep maintaining the dataset and include the evaluation result on this data in the final version.
>
> [**Evaluation of SOTA method**] We agree and provide additional evaluation results on four (*CRNN, ASTER, SVTR, ABINet*) SOTA methods tested on Latin and Chinese texts [ see table 3) in the revised main paper. We also provide the learning curve for these models on both the IID and OOD test data( see figure 12 in the revised supplementary material).
>
> [ **vector graphics**] to have an improved image quality, we utilize images in high-resolution .PDF format, except in cases where the original images are distorted.
>
>
> **Final remark**:  We would also like to thank you for reviewing our work and providing your valuable comments. We believe our responses have addressed your concerns regarding evaluation of additional SOTA methods and evaluation metrics. If so, we kindly request your consideration for a potential score adjustment.

---

> > ### Comment · Reviewer_BcAH · 2023-08-30
> >
> > Thanks for your responses and supplementary experiments. The responses have addressed most of my concerns and I'll raise my rating.

---

### Official Review · Reviewer_uxiq · 2023-07-20
**A Historical Handwritten Dataset for Ethiopic OCR**

**Rating:** 5
**Confidence:** 3

**Strengths:**

1.	The paper provides the first large dataset for recognizing historical handwritten text images from Ethiopia.
2.	The baselines are presented in groups by model structures and the reasons for differences in CERs are analyzed in detail.
3.	Human-level recognition results are provided to facilitate the evaluation of model performance.


**Additional Feedback:**

The manuscript lacks a detailed description of the sample characteristics and statistical information of the dataset. There is also a lack of results from testing datasets with existing OCR engines.

**Clarity:**

The writing could be improved in some areas to enhance the clarity and readability of the paper.

**Correctness:**

The dataset lacks detailed statistical information about the samples.
1) Class distribution. Unbalanced class distribution can cause hidden problems in the recognition system.
2) The size of the width of the text line image plays an important role in the recognition system. Therefore, the distribution of samples should be given based on the width of the text line image.
3) The number and percentage of each class in the training and test set.


**Documentation:**

The authors fully describe the collection and annotation process for this dataset. They also detail the model architecture used to train the baseline results.

**Ethics:**

no.

**Limitations:**

1. There is a lack of results from testing the dataset with existing OCR engines that support Amharic (e.g. Abyssinica OCR or Tesseract).
2. The test dataset II is assumed to have a different distribution than the training set. However, the images of documents from different centuries in Figure 1 look similar. It may not be correct to assume that the 18th century manuscript samples have different distributions.
3. The annotators for the IID subset and the OOD subset in Table 2 are not consistent, which may lead to unfair results for human-level performance.


**Opportunities For Improvement:**

1、	What categories of annotations does the dataset contain? Which tasks are they adapted to? Give examples of annotations.
2、	The manuscript lacks a detailed description of the characteristics of the dataset, such as the diversity of content, the diversity of image quality, and so on.
3、	Lack of detailed statistical information on the dataset such as 1) the class distribution as an unbalanced distribution can cause hidden problems in the recognition system. 2)The size of the width of the text-line images plays an important role in the recognition system. Therefore, the distribution of samples should be given based on the width of the text-line images. 3)The number and percentage of each category in the training and test sets.


**Relation To Prior Work:**

The paper does not explicitly discuss how this work differs from previous contributions.

**Summary And Contributions:**

The paper presents the largest Ethiopian OCR dataset to date, with a subset of OOD (Out-Of-Distribution) provided in its test set to facilitate evaluation of the model's generalization performance. The authors provide baseline results using a trained traditional CTC model and a model based on the Transformer architecture. The recognition accuracy of the OCR model is comparable to that of a human, suggesting that the dataset mitigates the difficulty of distinguishing between structurally similar Ethiopian scripts to some extent.

---

> ### Author Response · Authors · 2023-08-19
>
> # Response to reviewer [uxiq]
> We appreciate your valuable feedback. Thank you.
>
> [**Types of annotation and examples**] The annotation is text-line transcription (for handwritten text recognition task). This dataset consists of text-line images of handwritten Ethiopic script paired with corresponding text transcriptions as a ground truth.
> Examples are provided in Figure 14 of the supplementary material in section C.3. In this visualization, text-line images are presented on top, with corresponding annotations given beneath each image in blue color.
>
>
> [**detail description about the diversity of the data**]  The HDD-Ethiopic text-line images originate from 7 distinct books, covering cultural and religious related contents. In order to ensure privacy, the organization (ENALA) employed a random selection process, and organized pages based on their year of publication before providing them to us for this task. Then we extracted text-lines from this randomly assembled page collection. Sample pages from the collection, categorized as having good, medium, and poor quality, are provided in the revised supplementary material ( see appendix section C.4).
>
> [**detail statistics of the dataset  and image size**] we agree with the reviewer that this information is so important to better understand the dataset and it is true that the size of the image affects the recognition performance. We provide the details statistic about  width and height distribution of the images, number of characters per class (a  github link https://github.com/bdu-birhanu/HHD-Ethiopic/tree/main/Dataset/distribution_of_characters (see  Appendix section C.4, Figure 17, and 18 in the revised supplementary material).
>
> [**testing the existing OCR engine**] We agree that evaluating the data set with existing OCR engines that support Amharic would be beneficial for validating our dataset. In this regard, we have previously evaluated models that were proposed for Amharic script, as indicated in the first two methods of Table 3 in the main paper. However, evaluating Abyssinica OCR with our dataset poses a challenge due to its lack of open sourcing. This makes it hard to compare directly with other state-of-the-art text-line recognition models. We will assess Tesseract and incorporate the result in the final version.
>
> [**distribution in OOD and IID testset**]  we agree that the two documents in figure one look identical, however, there are significant differences to consider from the detailed statistics. For example color-bleeding and paper degradation is observed in the ood test set, while considerably less in the iid test set. Furthermore, the formatting, and layout in the iid test set have greater similarity to the training set compared to the OOD test set. Additionally, the 18th-century dataset contains longer texts, as evident in the histogram presented in  Appendix, section C.4, Figure 18 in the revised supplementary material.
>
> [**number of participants in Human-level performance evaluation**] we recognize varying participant counts in evaluating human-level performance. This wasn't intentional. Initially, we employed 10 annotators for each subset, but some didn't complete the annotation on time and resigned from the task. As a result, we utilized the annotations that were successfully submitted for evaluation. Further information, including detail process and justification, for both training and test data annotations, is given in Section C.3, L492-L532 of the supplementary material.
>
> [**unbalanced class distribution in rare characters**]  we agree with the reviewer and it is true that unbalanced distribution causes hidden problems in the recognition system. However it is common within writing systems and real datasets that some characters are frequently used while others are less common. We also acknowledge this limitation in the conclusion section of the main paper. As a solution, we also generated new artificial text-line images for characters which occur less frequently and added them to the original dataset in Hugging Face. (https://huggingface.co/datasets/OCR-Ethiopic/HHD-Ethiopic/tree/main/train/train_raw/under_represented_char_synth).
>
> [**language edition**] we made additional editing in many sections of the paper.
>
> [**contribution**] The main contribution of this paper is creating a dataset for a low-resourced script that has not been well investigated in OCR development, evaluating human-level performance and compared with SOTA OCR models  and these details were provided in the contribution section from L52-L62.
>
> **Final remark**:  Once again, we thank you for reviewing our work. We hope that our responses have addressed your questions concerning image size distribution, reasons behind human-level performance, and  evaluation of existing methods. If so, we kindly request your thoughtful consideration for a potential score adjustment.

---

> ### Comment · Reviewer_uxiq · 2023-08-22
>
> The author partially addressed my concerns. However, based on the current state of the paper and the author's response, I stand by my original score.

---

> > ### Author Response · Authors · 2023-08-22
> >
> > *Thank you for taking the time to review our revised paper and the responses provided for your comments. As we continue working on addressing the feedback, we kindly request the reviewer's additional guidance and any remaining concerns that might not have been covered in our previous response. Your insights are greatly valued as we work towards enhancing the quality of our paper.*

---

### Official Review · Reviewer_cn6S · 2023-07-20
**Obvious contribution but several shortcomings.**

**Rating:** 6
**Confidence:** 3
**Clarity:** This paper is well written.

**Strengths:**

This paper is well written, and the contribution is clearly demonstrated.This paper fill the gap of suitable datasets for machine learning tasks in historical handwritten Ethiopic text-image recognition. This research work will promote the research related to history and archaeology.

**Additional Feedback:**

Aims to fill the gap of suitable datasets for machine learning tasks in historical handwritten Ethiopic text-image recognition, this paper proposes a large dataset for historical handwritten Ethiopic text-image recognition, named HHD-Ethiopic. This dataset contains a training set and two test sets, one for classical IID setting and the other for a different distribution from the training set. In the experiments, the authors evaluate four approaches and two advanced versions on the proposed dataset and use human-level performance as a baseline. According to the quality of the paper, my comments are listed as follows:

1.	What are the concrete differences between the two test sets in the proposed dataset? For example, the OOD test set has color bleed-through or paper degradation, which is unseen in the training set, while the ID test set does not have these unseen cases.
2.	Are there any other datasets available for historical handwritten Ethiopic text-image recognition before? If so, the authors need to compare them with the proposed dataset in terms of the number of images, inclusion of more challenging cases and higher resolution, etc.
3.	I recommend the authors to vertically center the contents of the first two columns of table 3, which might make table 3 more aesthetically pleasing.
4.	Although the author has used the SOTA methods in 2023 for evaluation, the number of selected methods for experiments is inadequate, which make it difficult to determine whether the mentioned challenges are universal.
5.	This article mentions a shortcoming of this dataset at the end of the manuscript. I suggest that the authors illustrate the problem more specifically through the specific cases to better inspire future research.
6.	The authors only use CER as the only evaluation metric, which may not be comprehensive enough.
7.	The volume of the proposed dataset does not seem to be large enough, should it be called a “large” dataset?


**Correctness:**

I think the proposed dataset is constructed reasonablely, but the authors only use CER as the only evaluation metric, which may not be comprehensive enough.

**Documentation:**

The author provides a Github link and related documentation, and provides downloadable links on zenodo and Hugging Face.

**Limitations:**

According to the quality of the paper, my comments are listed as follows:

1.	What are the concrete differences between the two test sets in the proposed dataset? For example, the OOD test set has color bleed-through or paper degradation, which is unseen in the training set, while the ID test set does not have these unseen cases.
2.	Are there any other datasets available for historical handwritten Ethiopic text-image recognition before? If so, the authors need to compare them with the proposed dataset in terms of the number of images, inclusion of more challenging cases and higher resolution, etc.
3.	I recommend the authors to vertically center the contents of the first two columns of table 3, which might make table 3 more aesthetically pleasing.
4.	Although the author has used the SOTA methods in 2023 for evaluation, the number of selected methods for experiments is inadequate, which make it difficult to determine whether the mentioned challenges are universal.
5.	This article mentions a shortcoming of this dataset at the end of the manuscript. I suggest that the authors illustrate the problem more specifically through the specific cases to better inspire future research.
6.	The authors only use CER as the only evaluation metric, which may not be comprehensive enough.
7.	The volume of the proposed dataset does not seem to be large enough, should it be called a “large” dataset?

**Opportunities For Improvement:**

Although this dataset contains a large number of cases and can be used to evaluate whether the model has generalization to OOD data, the underrepresentation of rare characters reduces the contribution of this dataset.

**Relation To Prior Work:**

The comparison with previous related datasets should be more adequate.

**Summary And Contributions:**

Aims to fill the gap of suitable datasets for machine learning tasks in historical handwritten Ethiopic text-image recognition, this paper proposes a large dataset for historical handwritten Ethiopic text-image recognition, named HHD-Ethiopic. This dataset contains a training set and two test sets, one for classical IID setting and the other for a different distribution from the training set. In the experiments, the authors evaluate four approaches and two advanced versions on the proposed dataset and use human-level performance as a baseline.

---

> ### Author Response · Authors · 2023-08-19
>
> # Response to reviewer [cn6S]
>
> We would like to thank you  for  the comments, suggestions  proposed  for improving our dataset quality by the reviewer.
>
> [**Underrepresentation of rear characters**]  since the HHD-ethiopic dataset is a real dataset it is normal to have underrepresented character due to the nature of the writing system. To handle the issue of underrepresentation, we have generated synthetic text-line images with the rare characters. We provide the detail statistic about the distribution of sample character in each 306 classes( see the appendix section c.4 in the revised  supplementary material)  and added them in Hugging Face together with the original data: https://huggingface.co/datasets/OCR-Ethiopic/HHD-Ethiopic/tree/main/train/train_raw/under_represented_char_synth.  We will keep maintaining our dataset and provide addition results in the final version.
>
> [**OOD vs IID testsets**]  Thanks for pointing out some of the differences in these test sets. It is true that color bleeding and paper degradation are mostly observed in the 18th century set and, to a considerable extent, in the IID test set. In addition, formatting, and layout in the IID test set are more similar to the training set compared to the OOD test set. Furthermore, the 18th century set has longer texts than the IID set (refer to the histogram in Figure 18 of Appendix Section C.4 in the revised supplementary material).
>
> [**availability of previously published dataset for Ethiopic historical handwritten**] to the best of our knowledge, no historical handwritten dataset for Ethiopic exists. Our HHD-Ethiopic dataset is the first contribution for this case. However, we have include a comparison table of publicly available modern printed, modern handwritten and scene text Amharic datasets (see table 4 in section c.4 of the revised supplementary material)
>
> [**Table 3 format correction**] we vertically centered all the contents in the revised version
>
> [**additional SOTA methods*]  we evaluate  about four additional SOTA methods (including CRNN, ASTER, SVTR, and ABINet) that have been widely evaluated for Latin and Chinese scripts. Results are reported in Table 3 of the revised main paper and the learning curve for these models on both test sets is also provide ( see figure 12 in the revised supplementary material).
>
> [**more examples about shortcoming of the dataset**]: we provide the limitations with more examples in the revised version of the supplementary material (see section C.5 figure 20 in the supplementary material)
>
> [**more evaluation Metrics**]  we provide additional commonly used metric (https://arxiv.org/pdf/1909.07741.pdf ), Normalized Edit Distance (NED) ( see Table 2: human-level performance) and (Table 3: models’ performance in the revised main paper)
>
> [naming related to dataset size] we agree and we referred to it as the 'sizable dataset” in the revised version.
>
> [**comparison with previous dataset**] there is no publicly available historical handwritten dataset for Ethiopic that we could directly compare with. However we listed and compared other modern printed, modern handwritten  and scene text dataset of Amharic script as it strength our literature review part (see Table 4 in appendix section C.4 in the revised  supplementary material)
>
> **Final remark**: We would like to thank you once again for your review and valuable comments. We believe our responses have addressed your questions and concerns regarding the evaluation of additional state-of-the-art methods, supplementary evaluation metrics, as well as the detailed solution to handle the issue of underrepresented rare characters. If you find our responses satisfactory, we kindly request your consideration for a potential score adjustment.

---

### Official Review · Reviewer_yVsL · 2023-07-21
**historical handwritten dataset (80K samples) for Ethiopic OCR**

**Rating:** 5
**Confidence:** 5
**Correctness:** None
**Clarity:** Yes

**Strengths:**

The authors
1. Construct a novel dataset consisting of a historical handwritten dataset.

2. Describe some characteristics of the Ethiopic language.

3. Report human-level performance.

**Additional Feedback:**

None

**Documentation:**

None

**Opportunities For Improvement:**

Major comments:
1. The experimental setting seems inappropriate. Particularly, the selection of models for experiments is inappropriate.
For example, Plain-CTC and Attn-CTC are not state-of-the-art (SOTA) methods and are not widely-used models as far as I know.
In addition, there is no description for Attn-CTC.
Generally, the researchers in this field use SOTA methods validated in English or Chinese.
However, Plain-CTC and Attn-CTC are not validated models, and they are just models the first authors used in his/her previous works.
In the case of NomNaOCR, I cannot find the paper and cannot know whether the method is validated or not.
There are plenty of SOTA methods such as CRNN (CTC), ASTER/TRBA (attention decoder), ABINet/PARSeq (Well-known transformer-based models),  and widely-used repositories such as deep-text-recognition-benchmark, PaddleOCR, EasyOCR, and MMOCR.
They are validated in previous works, and thus I recommend using those methods.

2. In text recognition, when the training data is not enough, researchers usually generate synthetic data for training.
The previous works of the first author (Plain-CTC [6] and Attn-CTC [8]) also generated and used synthetic data.
Why the authors did not generate synthetic data for this time?

3. Performance of TrOCR and human-level performance is too low than my expectation. I wonder why.
Considering human-level performance is low, are there many noisy labels in label annotation?

4. If possible, it would be better to have an apparent reason for creating/using the Ethiopic dataset compared to other languages.
In other words, there are many datasets from many different languages in the world, and it would be better to clarify the reason for studying the recognition of Ethiopic languages and the differences from other datasets.
While the authors emphasized the difficulty of the Ethiopic language, I think texts from other languages, such as Chinese, Hindi, and Arabic, are much more difficult to recognize.


Minor comments:
1. The citations of L87-89 seem wrong “However, many of these applications have not been successful in recognizing text in historical manuscripts, particularly handwritten Ethiopic manuscripts [7, 8].” [7, 8] did not compare recent state-of-the-art methods and just used 1~3 methods mainly suggested by authors.

2. From Figure 4, why the color of the background of the samples is white? Are they different from the real data? Are they preprocessed data?

3. From Table 1, if the reason for OOD is because the test set consists of portion D from the 18th century, it seems a somewhat naive criterion.
Because the publication date for portion A is unknown, 18th century manuscripts may be included in A.
In addition, I wonder if there are apparent differences between 18th and 19th-century manuscripts.
For example, are manuscripts from the year 1790 and the year 1800 clearly different?

4. In my opinion, moving L160 “using a sklearn train/test split protocols from the training set.” into 3.1 would be better.

5. For evaluation metrics, why did the author not use normalized edit distance like ArT, LSVT, and ReCTS (ICDAR2019)?

6. L240 Table 2 -> Table 3

7. From L249 “benefiting from its exclusive training on historical documents.”, what does this mean?

**Relation To Prior Work:**

Yes

**Summary And Contributions:**

This work provides a historical handwritten dataset (80K samples) for Ethiopic OCR, named HHD-Ethiopic.
The authors evaluate OCR performance (line-level text recognition) on HHD-Ethiopic with several methods.
The average results are low. It may indicate the difficulty of the dataset.
The authors also report the human-level performance on the dataset.

---

> ### Author Response · Authors · 2023-08-19
>
> # Response to reviewer [yVsL]
>
> Thank you for dedicating your time and effort to reviewing our paper. Your feedback has helped us to enhance and refine our work. We address your concerns and questions as follows.
>
> [**SOTA Methods**] We agree with this comment, as it enhances our dataset's validity. Therefore, we evaluated four  additional SOTA methods  (*CRNN, ASTER, SVTR, ABINet*) as recommended and omitted the NomNaOCR (see table 3 in the revised main paper.) A learning curve  is also given in the revised supplementary material Figure 12.
>
> [**models description e.g Attn-CTC**] In this case  we provided detail settings for the plain-CTC and Attn-CTC methods, along with the architecture and training procedures in detail in Supplementary material section C.2.
>
> [**synthetic data**] We agree that training a model with large synthetic images could improve recognition performance. So, we generated synthetic data to add to our collection. We will keep our dataset and show more results in the final version.
>
> [**TrOCR and human level performance**]  We agree that the TrOCR and human-level performance is low and we discussed potential reasons in section C.2 of the supplementary material.
>
>  * For TrOCR, finetuning from a pre-trained model  for 3  epochs is done due to its size. In this training setting it does not outperform other models (Appendix, Section C.2).*
>
> *For human level performance, the individuals participating in human-level recognition performance are not experts in historical manuscripts. This lack of expertise could potentially contribute to lower prediction scores.*
>
> *Regrading the annotation details, we work to the possible maximum effort so as to have minimized noise labels in the ground truth by employing experts of historical manuscripts to double check the ground truth texts (see the details of the annotations in the supplementary material section C.3 from line 492-532)*
>
> [**motivation for creating the dataset**] the motivation behind creating the HHD-Ethiopic dataset for Ethiopic script is its classification as a low-resourced script and provided more details in the datasheet (section A) of the supplementary material.
> Creating a dataset for such a script has various benefits such as:
> - Advancing OCR technology for diverse scripts.
> - Digitizing and preserving historical handwritten Ethiopic texts to make them easy to edit and search.
> - Enhancing accessibility to historical knowledge and facilitating easy digital sharing.
>
> *We also recognize that certain scripts present greater recognition challenges than others. For instance, when evaluating character counts, the Ethiopic script consists of about 317 characters with visually similar forms compared to scripts like Arabic and Hindi (refer to a list of Ethiopic alphabets and level of complexities in Supplementary material section B).*
>
> [**errors in citation**]  We corrected and highlighted the revision.
>
> [**properties of sample images from the dataset**]  Sample text-line images shown in figure 4  are preprocessed binary images.
>
> [**OOD vs IID test sets**]  The differences are evident in various ways. For instance, the OOD test set comprises longer texts and degraded documents in comparison to the IID test set ( section C.4 Figure 15 and  18 see revised supplementary material). alternatively, as there is no cost involved for us or the OCR model, we could combine the test sets later to maintain this dataset (though it is recommended to use them separately for evaluation).
>
> [**Evaluation metrics**]  new reports are included, using NED metrics, in the revised version for both human-level and model performance (see Table 2 and 3 in the revised main paper)
>
> [**typo errors, content arrangement and ambiguous statements**] we have corrected the typographical error and reorganized the contents in Sections 3.1 and 3.2. We have also eliminated any ambiguous statements in the revised main paper.
>
> **Final remark**: We thank you again for reviewing our work and for the constructive comments. We hope that our responses have addressed your concerns about the SOTA methods, evaluation metrics, and  reason for non-inclusion of synthetic text-line images during training. If so, we would appreciate it if you could consider a potential score adjustment.

---

> > ### Comment · Reviewer_yVsL · 2023-08-31
> >
> > Thank you for your response. I raise my rating since most concerns seem addressed.
> >
> > In the case of TrOCR, how about removing TrOCR from the paper?
> > I don't know much about TrOCR, but I guess TrOCR requires more tuning for Ethiopic data.
> > For now, the authors have already used other SOTA methods, so TrOCR seems unnecessary.
> > It is just a comment; please ignore it if the authors do not want to.

---

### Author Response · Authors · 2023-08-19

## General comment

We thank the reviewers for their time and effort in reviewing our work. Their recognition of the dataset's significance and the importance of evaluating human-level performance is appreciated. The reviewers raised concerns about evaluation metrics and the need for more state-of-the-art (SOTA) methods.

We have addressed these concerns by conducting additional experiments, revising the paper, and including  figures and text. We trained additional SOTA methods and reported their recognition performance with additional  evaluation metrics. We believe these updates in the main paper, along with the detailed discussions in the supplementary material, enhance the clarity and validity of our work.

---

### Decision · Program_Chairs · 2023-09-22

**Decision:**

Reject

**Comment:**

This paper received four reviews with mixed opinions. Although two reviewers raised their ratings during the rebuttal period, two reviewers remained unconvinced, leaning towards the negative side. They remained unconvinced due to lack of comprehensive evaluation based on multiple metrics and baseline models. Moreover, the paper had several typos and mistakes. Based on the overall view, I recommend rejection.